# Unique transcriptional and protein-expression signature in human lung tissue-resident NK cells

Nicole Marquardt [1], Eliisa Kekäläinen[1,2,3], Puran Chen[1], Magda Lourda [1,4], Jennifer N. Wilson [1], Marlena Scharenberg [1], Per Bergman[5], Mamdoh Al-Ameri[5], Joanna Hård[6], Jeffrey E. Mold[6], Hans-Gustaf Ljunggren[1] & Jakob Michaëlsson[1]

Human lung tissue-resident NK cells (trNK cells) are likely to play an important role in host responses towards viral infections, inflammatory conditions and cancer. However, detailed insights into these cells are still largely lacking. Here we show, using RNA sequencing and flow cytometry-based analyses, that subsets of human lung CD69+CD16− NK cells display hallmarks of tissue-residency, including high expression of CD49a, CD103, and *ZNF683*, and reduced expression of *SELL*, *S1PR5*, and *KLF2/3*. CD49a+CD16− NK cells are functionally competent, and produce IFN-γ, TNF, MIP-1β, and GM-CSF. After stimulation with IL-15, they upregulate perforin, granzyme B, and Ki67 to a similar degree as CD49a−CD16− NK cells. Comparing datasets from trNK cells in human lung and bone marrow with tissue-resident memory CD8+ T cells identifies core genes co-regulated either by tissue-residency, cell-type or location. Together, our data indicate that human lung trNK cells have distinct features, likely regulating their function in barrier immunity.

[1] Center for Infectious Medicine, Department of Medicine, Karolinska Institutet, Karolinska University Hospital, Stockholm, Sweden. [2] Immunobiology Research Program & Department of Bacteriology and Immunology, University of Helsinki, Helsinki, Finland. [3] HUSLAB, Division of Clinical Microbiology, Helsinki University Hospital, Helsinki, Finland. [4] Childhood Cancer Research Unit, Department of Women's and Children's Health, Karolinska Institutet, Karolinska University Hospital, Stockholm, Sweden. [5] Thoracic Surgery, Department of Molecular Medicine and Surgery, Karolinska Institutet, Karolinska University Hospital, Stockholm, Sweden. [6] Department of Cell and Molecular Biology, Karolinska Institutet, Stockholm, Sweden. Correspondence and requests for materials should be addressed to N.M. (email: nicole.marquardt@ki.se)

Mucosal surfaces in the human lung are frequently exposed to microbes, environmental particles, and other forms of antigens. Consequently, the lung is often subject to severe acute and chronic inflammatory diseases and malignancies exemplified by allergies, asthma, microbial infections, and cancer, many of which are major causes of morbidity and mortality. Tissue-resident lymphocytes that are located close to the airways in the lung are exposed to many of these potentially disease-causing agents and/or malignant cells. Depending on their functional responses, they likely play important roles in the pathogenesis of many of the above-mentioned diseases.

Human lung tissue-resident memory $CD8^+$ T ($T_{RM}$) cells have been demonstrated to have a unique transcriptional profile[1,2]. Furthermore, the frequency of $T_{RM}$ cells in tumors have been shown to correlate positively with clinical outcome in lung cancer[3,4]. Of note, exploitation of $T_{RM}$ cells is currently discussed as a potential approach in human cancer immunotherapy[5]. In contrast to $T_{RM}$ cells, surprisingly little is known about tissue-resident NK (trNK) cells in human lung. Indeed, most of our knowledge about human NK cells is still largely deduced from ex vivo studies of NK cells derived from peripheral blood. NK cells constitute an important part of the innate immune defense with the capacity to kill virus-infected and malignant cells and mediate immunoregulatory responses by releasing several proinflammatory cytokines such as IFN-γ and TNF[6]. With respect to human lung NK cells, they have been phenotypically characterized to some extent and shown to be hyporesponsive toward tumor target cell stimulation ex vivo[7]. They do, however, respond to virus-infected target cells[8], indicating that lung trNK cells could be clinically relevant in pathological conditions.

Approximately 10% of $CD56^{dim}CD16^+$ NK cells and 75% of $CD56^{bright}CD16^-$ NK cells in the human lung express CD69[7]. CD69 has been demonstrated to inhibit the egress of T cells from tissues by interfering with the sphingosine-1-phosphate receptor ($S1P_1$)[9], and is therefore considered a hallmark marker of tissue-resident lymphocytes. In addition, the integrins CD49a (α1 integrin) and CD103 (αE integrin) are involved in retaining lymphocytes in tissues and, hence, identify $T_{RM}$ cells and trNK cells[10–12]. CD49a binds to collagen IV, which is a main component of the basement membrane of the blood–gas–barrier in the lung but is largely absent in the lung parenchyma. CD103 binds to E-cadherin, which is expressed by epithelial cells lining the alveolar spaces and bronchial epithelium of the lung, as well as on lung tumors of epithelial origin[13]. $CD103^+$ $CD8^+$ $T_{RM}$ cells have been demonstrated to be located close to the epithelia in the human lung[14], hence, the expression of CD103 on tissue-resident lymphocytes is indicative of their localization. Distinct tissue-specific expression patterns of CD49a and CD103 have been detected on NK cells in human liver[15], tonsil[16], and endometrium[17], indicating that there are discrete subsets of trNK cells in each of these organs. Moreover, recent studies of $CD49a^+$ NK cells in human liver have revealed subsets of $Eomes^-T-bet^+$ and $Eomes^+T-bet^{low}$ NK cells, identifying potentially distinct lineages of $CD49a^+$ trNK cells within this organ[15,18,19]. In contrast, comparatively much less is known about the nature of human trNK cells in many other tissues including the lung.

Here, we analyze phenotypic, transcriptional, and functional traits of discrete NK cell subsets in the human lung with respect to tissue-residency. We identify NK cell subsets with a gene and protein-expression profile consistent with that of tissue-resident lymphocytes, e.g., cell surface expression of CD69, CD49a, and CD103. Comparisons of gene expression patterns in NK cells and $CD8^+$ T cells in different tissues reveal core genes co-regulated either by tissue-residency, cell-type or location. Together, our data demonstrate that $CD69^+$ NK cells co-expressing CD49a and CD103 display distinct hallmarks of tissue-residency with a tissue-specific and functional signature. These findings contribute to the understanding of lung tissue-associated barrier functions as well as the role of trNK cells in lung pathologies, such as viral infections and cancer.

## Results

**An NK cell subset in human lung expresses CD49a and CD103.** The majority of $CD56^{bright}CD16^-$ NK cells and a subset of $CD56^{dim}CD16^+$ NK cells in human lung expresses CD69[7], a marker commonly expressed by tissue-resident lymphocytes. In order to determine whether additional markers of tissue-residency were expressed on human lung $CD69^+$ NK cells, we analyzed expression of the integrins CD49a and CD103 on these cells (Fig. 1). NK cells were identified as live $CD45^+CD3^-CD14^-$ $CD19^-CD56^+$ lymphocytes (Supplementary Fig. 1a). Both CD49a and CD103 were expressed on a significant proportion of $CD16^-CD69^+$ NK cells and to a lesser degree on $CD16^+CD69^+$ lung NK cells (Fig. 1a, b). In contrast, these molecules were detected at very low frequency on NK cells isolated from peripheral blood (Supplementary Fig. 1b). Furthermore, CD49a and CD103 were commonly co-expressed (Fig. 1b, c), although subsets of $CD49a^+CD103^-CD69^+$ NK cells and, to a lesser extent, $CD49a^-CD103^+CD69^+$ NK cells were detected (Fig. 1b), in line with a recent report[8]. The vast majority of $CD69^+CD49a^+$ $CD103^-$, $CD69^+CD49a^-CD103^+$, and $CD69^+CD49a^+CD103^+$ NK cells lacked expression of CD16 (Fig. 1d, e). Conversely, the vast majority of the $CD49a^+CD103^-CD16^-$, $CD49a^-CD103^+$ $CD16^-$, and $CD49a^+CD103^+CD16^-$ NK cells co-expressed CD69, whereas only approximately 50% of $CD49a^-CD103^-$ $CD16^-$ NK cells co-expressed CD69 (Fig. 1f). However, in a few donors CD49a and CD103 were co-expressed at a higher frequency also on $CD69^+CD16^+$ lung NK cells (Fig. 1b). In these donors, CD49a and CD103 were expressed predominantly by a subset of $CD56^{bright}CD16^-$ NK cells (Supplementary Fig. 1d, e). Furthermore, a subset of $CD56^{dim}CD16^-$ NK cells accounted for another minor population of lung NK cells expressing CD49a and CD103 (Fig. S1d, e). On average, the $CD69^+CD49a^+CD103^-$ $CD16^-$, $CD69^+CD49a^-CD103^+CD16^-$, and $CD69^+CD49a^+$ $CD103^+CD16^-$ NK cells together made up 20% of the total NK cell population in lung. $CD16^+CD69^-$ NK cells made up the majority of NK cells in the human lung, as previously reported[7] (Fig. 1g). The total frequencies of $CD69^+CD49a^+$ $CD103^-CD16^-$, $CD69^+CD49a^-CD103^+CD16^-$, and $CD69^+$ $CD49a^+CD103^+CD16^-$ NK cells detected were independent of clinical parameters, including medication and type of cancer of patients from whom clinical samples were derived (Supplementary Fig. 2a–c). However, frequencies of these cells were higher in the lungs of active smokers as compared to ex-smokers and non-smokers (Supplementary Fig. 2d). Noteworthy, non-NK innate lymphoid cells (ILCs), here defined as live $CD45^+CD14^-$ $CD19^-CD3^-CD16^-NKG2A^-KIR^-CD69^+CD127^+$ lymphocytes (Supplementary Fig. 1f), expressed substantially less CD49a and CD103 compared to NK cells and $CD8^+$ T cells in the human lung (Fig. 1h, i). Since $CD103^+CD8^+$ $T_{RM}$ cells have previously been shown to be located in or close to the lung epithelium where ligands for both CD103 (E-cadherin) and CD49a (collagen) are expressed, our data suggests that $CD49a^+CD103^+$ NK cells have an intraepithelial location. In summary, we show that a substantial fraction of primarily $CD16^-$ NK cells co-express CD69, CD49a, and CD103, and thus may represent trNK cells in human lung.

**Lung trNK cells segregate on protein and transcription level.** To characterize the subsets of $CD16^-$ NK cells in human lung, we next analyzed protein expression on $CD16^-$ NK cell subsets

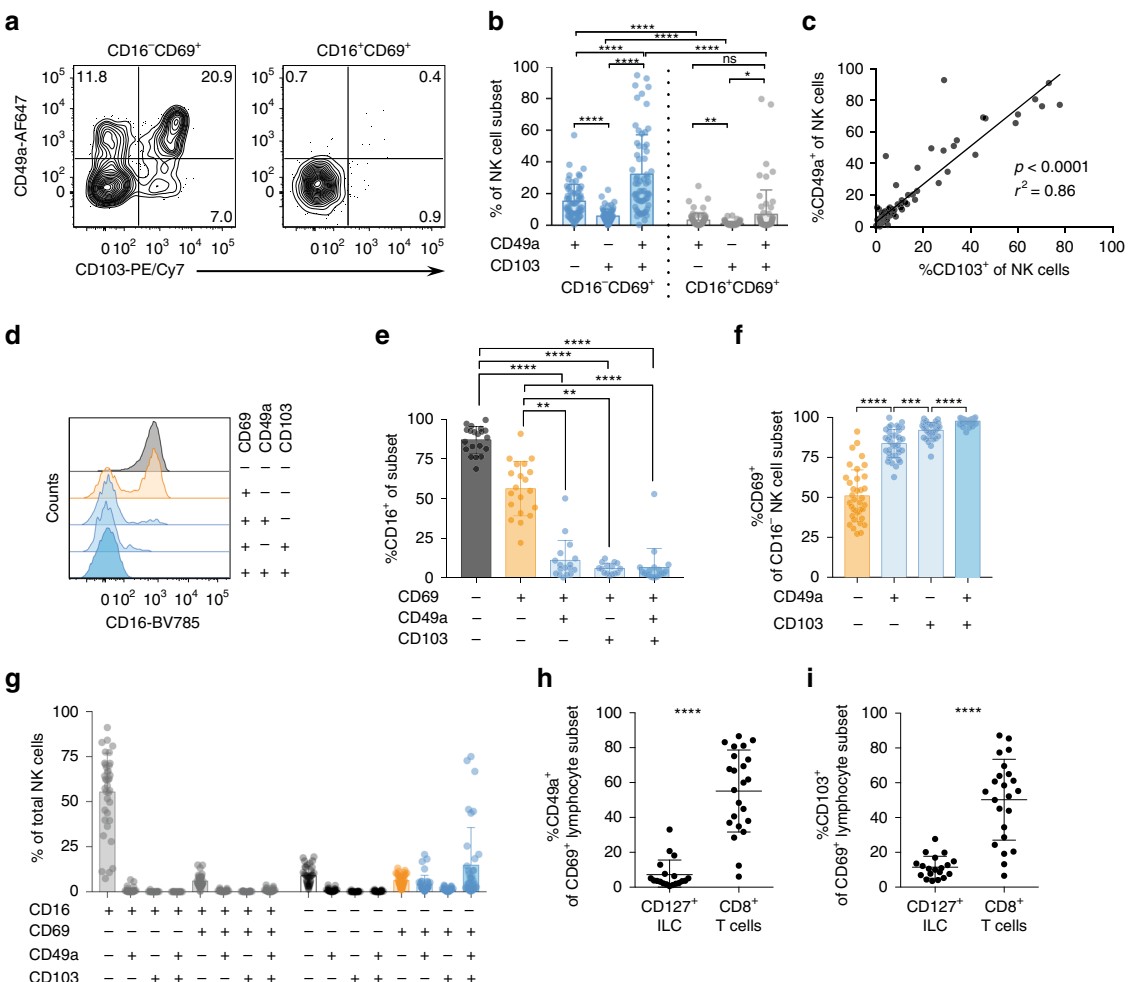

**Fig. 1** CD69+CD16− NK cells in the human lung co-express CD49a and CD103. **a** Representative dot plots and **b** summary of data of CD49a and CD103 expression on CD69+CD16− and CD69+CD16+ NK cells (n = 64). Mean ± SD is shown. **c** Linear regression analysis of CD49a and CD103 co-expression on NK cells in human lung (n = 98). **d** Representative histogram and **e** summary of data of CD16 expression on CD69− (n = 20), CD69sp (n = 20), and CD69+CD49a+CD103− (n = 17), CD69+CD49a−CD103+ (n = 15) and CD69+CD49a+CD103+ (n = 19) NK cells in human lung. Mean ± SD is shown. **f** Summary of data for CD69 expression on human lung CD49a−CD103−CD16− (orange), CD49a+CD103−CD16−, CD49a−CD103+CD16−, and CD49a+CD103+CD16− (all blue) NK cells (n = 37). Mean ± SD is shown. **g** Frequency of NK cell subsets defined by the expression of CD16, CD69, CD49a, and CD103 out of total NK cells (n = 37). Mean ± SD is shown **h** Percentages of CD49a+ and **i** CD103+ cells among CD127+ ILCs (defined as live CD45+CD14−CD19−CD3−CD16−NKG2A−KIR−CD69+CD127+) (n = 19) and CD69+CD8+ T cells in the human lung (n = 23). Mean ± SD is shown. Statistical analysis in **b**, **f** RM one-way ANOVA, Tukey's multiple comparison test, Geisser-Greenhouse's correction. *p < 0.05, **p < 0.005, ***p < 0.001, ****p < 0.0001; ns, non-significant; **e** Kruskall-Wallis test, Dunn's multiple comparisons test. **p < 0.005, ****p < 0.0001; **h**, **i** Two-tailed Wilcoxon matched-pairs signed rank test, ****p < 0.0001. Source data are provided as a Source Data file

defined by distinct patterns of expression of CD69, CD49a, and CD103 (Fig. 2a–h). CD69+CD49a−CD103−CD16− (CD69sp), CD69+CD49a+CD103−CD16−, and CD69+CD49a+CD103+ CD16− NK cells almost completely lacked expression of CD57 and expressed high levels of NKG2A (Fig. 2a, b). We however noted that a fraction of the CD69−CD16− NK cell subset expressed CD57 (Fig. 2b), indicating that this subset might include CD56dimCD16+ NK cells that had downregulated expression of CD16. In the subsequent analyses, we therefore gated on CD57−NKG2A+ cells among the CD69−CD16− NK cells (hereafter referred to as "CD69−CD16−" NK cells) to avoid including CD56dim NK cells that had lost CD16 expression. As expected, all CD16− NK cell subsets expressed higher levels of CD56 compared to CD16+ NK cells (Fig. 2c, d). Next, we analyzed the expression of the chemokine receptors CXCR3 and CXCR6, which have previously been shown to be important for

tissue-homing (Fig. 2c, d). While CXCR3 was highly expressed on CD69−CD16−, CD69+CD49a+CD103−CD16−, and CD69+ CD49a+CD103+CD16− NK cells, levels were low on the CD69spCD16− and CD56dimCD16+ NK cells. In contrast, CXCR6 expression was modestly, but yet significantly, increased on CD69+CD49a+CD103+CD16− compared to CD69−CD16− NK cells (Fig. 2c, d). These results indicate potential differences in the tissue-homing capacity between the NK cell subsets. Ex vivo, all CD69+CD16− NK cell subsets analyzed (CD69sp, CD69+ CD49a+CD103− and CD69+CD49a+CD103+) expressed perforin, albeit at lower levels than CD16+ NK cells (Fig. 2e, f). Finally, both the CD69− and CD69+ NK cell subsets expressed Eomes, and, at lower levels, T-bet, identifying them as bona fide NK cells (Fig. 2g, h).

To further characterize the subsets of CD16− NK cells, we performed RNAseq analysis of four subsets of CD56+NKG2A+

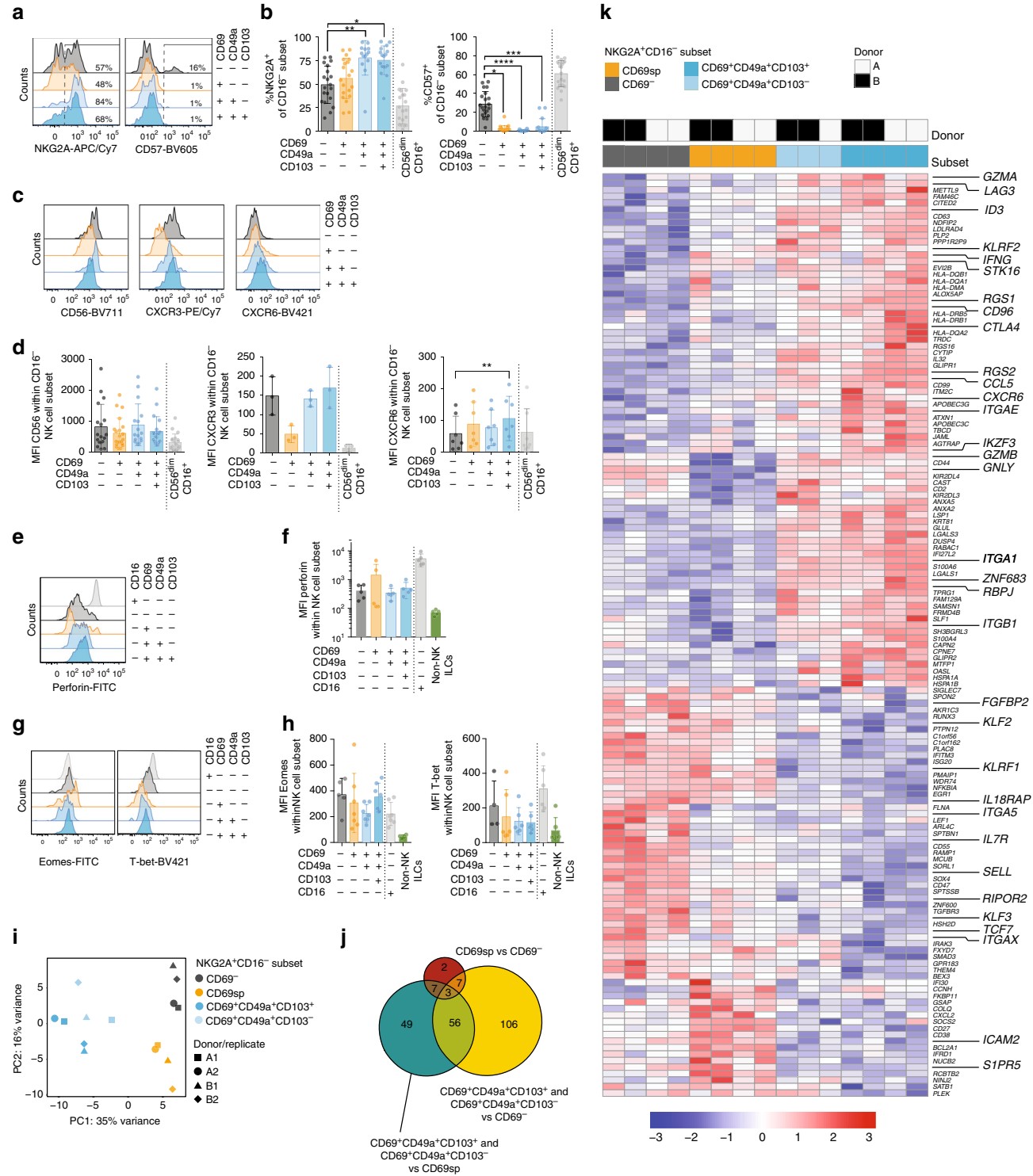

CD16− NK cells from human lung: CD69−, CD69+CD103−CD49a− (CD69sp), CD69+CD49a+CD103−, and CD69+CD49a+CD103+ NK cells (Fig. 2i–k, see Supplementary Fig. 1g for gating strategy). In addition, we analyzed CD56dimCD16+NKG2A+CD57− NK cells from the human lung as a reference population (see Supplementary Fig. 1g for gating strategy). Principal component analysis (PCA) revealed that the CD69+CD49a+CD103− and the CD69+CD49a+CD103+ subsets were clearly distinct from the CD69sp and CD69− subsets, as indicated by separation along PC1 (Fig. 2i). Furthermore, the CD69sp and the CD69− subsets were separated along PC2 (Fig. 2i), showing that also these populations were distinct from each other. As a reference, CD56dimCD16+NKG2A+CD57− NK cells were included in a separate PCA (Supplementary Fig. 3a). The latter NK cells clearly separated from all CD16− subsets, in line with previous studies comparing CD56dim and CD56bright NK cells[20–22].

Next, we analyzed differentially expressed genes between the four different subsets. In this analysis, we identified 230 differentially expressed genes (adjusted p-value < 0.05, and more

**Fig. 2** Human CD69⁺CD16⁻ lung NK cell subsets have unique characteristics. **a** Representative overlays and **b** summary of data of expression of NKG2A and CD57 on CD16⁻ human lung CD69⁻CD49a⁻CD103⁻ ($n = 20$ for both), CD69sp ($n = 20$ for both), CD69⁺CD49a⁺CD103⁻ ($n = 14$ and $n = 13$, respectively), and CD69⁺CD49a⁺CD103⁺ ($n = 17$ and $n = 15$, respectively) NK cells. CD56ᵈⁱᵐCD16⁺ NK cells are shown for comparison ($n = 20$ and $n = 19$, respectively). Numbers in **a** indicate %NKG2A⁺ and %CD57⁺ NK cells, respectively. **c** Representative overlays and **d** summary of mean fluorescence intensity (MFI) of CD56 ($n = 18$ for CD16⁻CD69⁻CD49a⁻CD103⁻NKG2A⁺CD57⁻, $n = 21$ for CD69⁺CD49a⁻CD103⁻CD16⁻, $n = 16$ for CD69⁺CD49a⁺CD103⁻CD16⁻, $n = 19$ for CD69⁺CD49a⁺CD103⁺CD16⁻, and $n = 20$ for CD56ᵈⁱᵐCD16⁺), CXCR3 ($n = 3$) and CXCR6 ($n = 7$) on human lung CD69⁻CD49a⁻CD103⁻NKG2A⁺CD57⁻, CD69sp, CD69⁺CD49a⁺CD103⁻, and CD69⁺CD49a⁺CD103⁺ NK cells within the CD16⁻ subset. **e** Representative overlay and **f** summary of MFI of perforin expression in CD16⁻CD69⁻CD49a⁻CD103⁻NKG2A⁺CD57⁻, CD69spCD16⁻, CD69⁺CD49a⁺CD103⁻CD16⁻, CD69⁺CD49a⁺CD103⁺ CD16⁻, and CD69⁻CD49a⁻CD103⁻CD16⁺ NK cells in human lung ($n = 5$). CD127⁺CD161⁺ cells from lung tissue are shown as a comparison ($n = 5$). **g** Representative overlays and **h** summary of MFI of Eomes (left panel) (CD16⁻CD69⁻CD49a⁻CD103⁻NKG2A⁺CD57⁻ ($n = 5$, other subsets $n = 7$) and T-bet (right panel) (CD16⁻CD69⁻CD49a⁻CD103⁻NKG2A⁺CD57⁻ ($n = 4$, other subsets $n = 6$) expression in CD16⁻ human lung CD69⁻CD49a⁻CD103⁻NKG2A⁺CD57⁻, CD69sp, CD69⁺CD49a⁺CD103⁻, and CD69⁺CD49a⁺CD103⁺ NK cells. Expression in CD16⁺CD69⁻CD49a⁻CD103⁻ NK cells and non-NK cell ILCs (CD56⁻CD16⁻NKG2A⁻KIR⁻CD161⁺) are shown as a reference for Eomes ($n = 4$), and T-bet ($n = 6$), respectively. **i** Principal component analysis based on 1,000 most variable genes across subsets. **j** Euler diagram based on the 230 differentially expressed genes (DEGs, $p < 0.05$, log2 fold change >1) between CD69⁺CD49a⁺CD103⁺NKG2A⁺CD16⁻, CD69⁺CD49a⁺CD103⁻NKG2A⁺CD16⁻, CD69spNKG2A⁺CD16⁻ and CD69⁻NKG2A⁺CD16⁻ NK cells. DEGs among CD69⁺CD49a⁺CD103⁺NKG2A⁺CD16⁻ and CD69⁺CD49a⁺CD103⁻NKG2A⁺CD16⁻ were pooled in the analysis. **k** Heatmap of 142 DEGs (padj < 0.01, log2 fold change >1) with genes ordered by hierarchical clustering. Genes involved in regulation of tissue-residency, homing, and/or NK cell function are highlighted. Two replicates per donor and subsets are shown. Significant DEGs identified using deSeq2. Colour scale depicts z-score. **b, d, f, h** Kruskal-Wallis test, Dunn's multiple comparisons test, *$p < 0.05$, **$p < 0.01$, ***$p < 0.001$, ****$p < 0.0001$. Means ± SD are shown. Source data are provided as a Source Data file

than one-fold log2 difference) between the different subsets (Fig. 2j, k, and Supplementary data 1). The largest number of differentially expressed genes was identified when comparing the CD69⁺CD49a⁺CD103⁻ and CD69⁺CD49a⁺CD103⁺ subsets with the CD69⁻ subset. In contrast, only a limited number of genes were differentially expressed between the CD69⁺CD49a⁺CD103⁻ and CD69⁺CD49a⁺CD103⁺ subsets, and between the CD69sp and CD69⁻ subsets, respectively (Fig. 2j, k, Supplementary Fig. 3b, and Supplementary data 1). Both CD69⁺CD49a⁺CD103⁻ and CD69⁺CD49a⁺CD103⁺ subsets of NK cells expressed higher levels of *ITGA1* (CD49a), *ZNF683* (Hobit), *RGS1*, and *RGS2*, and lower levels of *SELL* (CD62L), *S1PR5*, and *KLF3* transcripts compared to the CD69⁻ subset (Fig. 2k). All of these genes have previously been described as hallmarks of CD8⁺ T_RM cells and have been shown to be involved in tissue-retention[23]. Furthermore, the CD69⁺CD49a⁺CD103⁺ NK cell subset expressed significantly higher levels of *ITGAE* (CD103), *ITGB1* (CD29), and *CXCR6*, and lower levels of *KLF2* compared to the CD69⁻ subset (Fig. 2k and Supplementary Fig. 3b). Notably, the CD69⁺CD49a⁺CD103⁺ subset only differed from the CD69⁺CD49a⁺CD103⁻ subset with lower expression of a handful of genes, including *SELL*, *SATB1* and *HSPA1A* (Supplementary Fig. 3b). Taken together, the gene expression signatures of CD69⁺CD49a⁺CD103⁻ and CD69⁺CD49a⁺CD103⁺ NK cells strongly indicate that they represent trNK cells.

Since CD69 is commonly considered a hallmark marker of tissue-resident lymphocytes, we additionally analyzed whether the CD69sp subset in the lung also expressed genes associated with tissue-residency despite the lack of CD49a and CD103. Relatively few differentially expressed genes were, however, detected when comparing the CD69sp and CD69⁻ subsets with each other (Fig. 2j, k and Supplementary Fig. 3b), although several of them have been associated with tissue-residency, such as *SELL*, *RGS1*, and *ALOX5AP*. In contrast, a substantially larger number of differentially expressed genes were detected when comparing the CD69sp and CD69⁺CD49a⁺CD103⁺ subsets, e.g., lower expression of *RGS1*, *RGS2*, and *ZNF683*, and higher expression of *SELL*, *KLF2*, and *S1PR5* in the former subset (Fig. 2j, k and Supplementary Fig. 3b). This indicates that CD69spCD16⁻NK cells in the human lung have a less distinct tissue-resident phenotype.

In addition to the identification of genes directly involved in tissue-retention, a number of genes involved in regulating NK cell function were also differentially expressed in the CD69⁺CD49a⁺CD103⁻ and CD69⁺CD49a⁺CD103⁺ NK cell subsets compared to the CD69⁻subset, including higher expression of *KLRF2* (NKp65), *CD96*, *LAG3*, and *CTLA4* and lower expression of *KLRF1* (NKp80), *IL18R1*, *IL18RAP*, and *IL7R* (CD127) in the former subsets (Fig. 2k, Supplementary Fig. 3b, and Supplementary data 1). Of note, the CD69sp subset differed from the CD69⁻, CD69⁺CD49a⁺CD103⁻, and CD69⁺CD49a⁺CD103⁺ subsets with lower expression of *GNLY* (Granulysin) and *GZMB* (Granzyme B) (Fig. 2k, Supplementary Fig. 3b), indicating that CD69spCD16⁻NK cells have reduced cytotoxic capacity.

Transcription factors control the expression of a large number of genes, and therefore differences in expression of transcription factors are likely to reflect important differences in the biology of NK cells. The CD69⁺CD49a⁺CD103⁻ and CD69⁺CD49a⁺CD103⁺ subsets indeed differed in their expression profile of a number of transcription factors compared to the CD69⁻subset (Fig. 2k and Supplementary Fig. 3b), further strengthening the notion that these cells constitute a distinct population of cells. For example, the CD69⁺CD49a⁺CD103⁻and CD69⁺CD49a⁺CD103⁺ subsets had increased expression of *ID3*, *STK16*, and *RBPJ*, and decreased expression of *KLF3*, *SOX4*, *EGR1*, and *TCF7* compared to the CD69⁻ NK cell subset (Fig. 2k and Supplementary Fig. 3b). Moreover, the CD69⁺CD49a⁺CD103⁺ subset expressed higher levels of the transcription factors *ZNF331*, *IKZF3*, *CITED2*, and *IRF4*, and lower levels of *KLF2*, *SMAD3*, *SATB1*, *RUNX3*, *MLX*, *ZNF600*, and *BACH2* compared to the CD69⁻subset (Fig. 2k, Supplementary Fig. 3b, and supplementary data 1).

Taken together, the combined gene and protein expression analyses showed that CD69⁺CD49a⁺CD103⁻CD16⁻ and CD69⁺CD49a⁺CD103⁺CD16⁻ NK cells were distinct from both CD69spCD16⁻ and CD69⁻CD16⁻ NK cells, and had a transcriptional signature indicative of tissue-residency in human lung. In contrast, CD69spCD16⁻ NK cells in human lung were overall more similar to CD69⁻CD16⁻ NK cells and, to a lesser extent, expressed genes associated with tissue-residency.

**TrNK cells differ between tissues and from CD8⁺ TRM cells.** Recently, the transcriptome of trNK cells in human bone marrow was published[20] and was found to share a transcriptional signature with human spleen CD8⁺ T_RM cells[1]. It however remains

unknown whether trNK cells are unique for their respective tissue environment, and to what extent they share a common transcriptional signature. We therefore next compared the profile of differentially expressed genes in lung trNK cells (CD69+CD49a+ CD103+CD16− trNK cells versus CD69−CD16− NK cells), with available published data on bone marrow trNK cells (CXCR6+ CD69+CD54+ trNK cells versus CXCR6−CD56bright NK cells)[20], as well as with CD8+ $T_{RM}$ cells in human lung and spleen (CD69+CD45RA−CCR7− $T_{RM}$ cells versus CD69− CD45RA−CCR7−CD8+ $T_{EM}$ cells)[1]. Out of the differentially expressed genes detected in trNK cells in lung (padj < 0.05), a substantial number were also differentially expressed in trNK cells in bone marrow (Fig. 3a), as well as in CD8+ $T_{RM}$ cells in lung (Fig. 3b). In contrast, substantially fewer genes were commonly differentially expressed (i.e., co-regulated) in trNK cells in lung and CD8+ $T_{RM}$ cells in spleen (Fig. 3c). The analysis also revealed a number of genes that were differentially expressed in opposite direction in lung and bone marrow trNK cells, including S1PR5, SMAD3 (↓ in lung, ↑ in bone marrow), ZNF683, CD82, ANXA2, and several HLA class II-encoding genes (↑ in lung, ↓ in bone marrow) (Fig. 3a).

To further investigate to what extent gene expression patterns were shared or differed across trNK cells in lung, trNK cells in bone marrow, CD8+ $T_{RM}$ cells in lung and CD8+ $T_{RM}$ cells in spleen, we selected genes that were differentially expressed in trNK cells in lung and in at least one of the other cell types. Out of the differentially expressed genes (padj < 0.05, log2 fold change > 1) in CD69+CD49a+CD103+CD16− trNK cells in lung, 102 genes were also found to be differentially expressed in trNK cells in bone marrow, CD8+ $T_{RM}$ cells in lung, and/or CD8+ $T_{RM}$ cells in spleen (Fig. 3d). Eight genes were differentially expressed across all trNK cells and CD8+ $T_{RM}$ cells, compared to their non-tissue-resident counterparts: SELL, ITGAE, CXCR6, RGS1, KLF3, FGFBP2, RAP1GAP2, and RIPOR2 (Fig. 3d, highlighted with red text). In addition, 35 of the differentially expressed genes in trNK cells in lung were also differentially expressed in the same direction in trNK cells in bone marrow, but not in any of the CD8+ $T_{RM}$ cell subsets (Fig. 3d, highlighted with blue text); e.g., CCL5, IKZF3, GPR183, IL7R, IL18RAP, IL18R1, ITGAX, and CD55. Conversely, trNK cells and CD8+ $T_{RM}$ cells in lung shared 23 differentially expressed genes that were not differentially expressed in the same direction in trNK cells in bone marrow; e.g., ZNF683, KLF2, S1PR5, KLRF1, RBPJ, SAMSN1, and TGFBR3 (Fig. 3d, highlighted with green text).

In addition to the comparisons above, we also compared the gene expression signature of CD69spCD16− NK cells in lung with that of trNK cells in bone marrow (Supplementary Fig. 4). Indeed, most of the differentially expressed genes in CD69spCD16− NK cells in lung, albeit not many, were also found to be differentially expressed in trNK cells in bone marrow (Supplementary Fig. 4a). However, CD69spCD16− NK cells in lung differed in their expression pattern of a number of genes associated with tissue-residency, e.g., S1PR5, KLF2/3, ZNF683, RGS1, and RIPOR2, compared to CD69+CD49a+CD103−CD16− and CD69+CD49a+CD103+CD16− NK cells (Supplementary Fig. 4b), indicating that they have a less distinct tissue-resident phenotype.

Taken together, the comparisons revealed a set of core-genes that identify but also segregated trNK cells and CD8+ $T_{RM}$ cells across tissues and cell types, and further highlighted gene expression patterns shared between trNK cells between tissues. These findings suggest that gene expression signatures in tissue-resident lymphocyte subsets are imprinted by the local environments in the respective tissues as well as being dictated by cell type.

**Stimulated lung trNK cells upregulate effector functions**. We have previously shown that human lung NK cells are hypofunctional in response to target cell stimulation, e.g., with low degranulation capacity. However, lung NK cells do exhibit a clear capacity to produce cytokines[7]. We here set out to extend this functional analysis to the level of specific subsets of trNK cells in the human lung. We first show that human lung CD69+ CD49a+ CD103−CD16− and CD69+CD49a+CD103+CD16− NK cells are functionally equipped at a transcriptional level with respect to expression of LAMP1, IFNG, TNFA, CCL3, CCL4, CCL5, and CSF2 (Fig. 4a). Notably, both CD69+CD49a+ CD103−CD16−and CD69+CD49a+CD103+CD16− NK cells expressed higher levels of CCL5 in comparison to CD69−CD16−, CD69spCD16−, and CD56dimCD57−NKG2A+ NK cells (Fig. 4a), suggesting a specific role for human lung trNK cells in production of CCL5 (RANTES) in the human lung. Furthermore, we identified elevated CCL3 (MIP-1α) and CSF-2 (GM-CSF) gene expression levels in CD69spCD16−lung NK cells (Fig. 4a).

At the protein level, CD49a+ lung NK cells (representing the majority of NK cells with a tissue-resident phenotype in lung) displayed similar functional capacities upon stimulation with PMA/ionomycin as CD49a− NK cells, albeit with elevated observed production of GM-CSF (Fig. 4b, c). Further analysis of sorted CD16− NK cells revealed that CD49a+CD103+CD16− NK cells degranulated less and produced less GM-CSF as compared to CD49a+CD103−CD16− NK cells, whereas TNF production was similar between the compared subsets (Fig. 4d). These data thus suggest that functional differences exist even among NK cells with a tissue-resident phenotype.

IL-15 is a cytokine crucial for development, maintenance and function of NK cells and $T_{RM}$ cells[10,24–26]. It is produced at high levels by dendritic cells[27], epithelial cells, and alveolar macrophages[28–30] following viral infections in the lung. Given the importance of IL-15 for NK cell and $T_{RM}$ cell function, as well as the potential importance of trNK cells in anti-viral responses in the lung, we also analyzed different subsets of NK cells in human lung with respect to responses to IL-15. Stimulation of sorted CD49a+CD16−, CD49a−CD16−, and CD16+ lung NK cell subsets with IL-15 for three days resulted in increased proliferation as indicated by elevated Ki67 expression in both CD49a+CD16− and CD49a−CD16− NK cells as compared to CD16+ NK cells (Fig. 4e, f). Furthermore, both CD49a−CD16− NK cells and CD49a+CD16− NK cells strongly upregulated expression of perforin and granzyme B following three days of stimulation with IL-15 (Fig. 4e, f). Notably, these effector molecules were low or virtually undetectable, respectively, following ex vivo analysis of the cells (Figs. 2f and 4e, f). In contrast, CCL5 was expressed in approximately 50% of CD16+ and of CD49a+CD16− NK cells even in a resting state, whereas CD49a−CD16−NK cells largely lacked CCL5 expression ex vivo. However, the CD49a−CD16−NK cells upregulated CCL5 expression to levels similar to that of CD49a+CD16− NK cells upon in vitro stimulation with IL-15 (Fig. 4e, f). While CD16+ NK cells were highly polyfunctional but poorly proliferative, CD49a+CD16− NK cells made up the dominant polyfunctional subset following stimulation with IL-15 (Fig. 4g). In contrast, CD49a−CD16− NK cells were oligofunctional and poised towards perforin production.

Stimulation with IL-15 also induced de novo CD49a expression on sorted CD49a−CD16− NK cells derived from the human lung (Supplementary Fig. 5a), in line with a recent report of CD49a−induction on peripheral blood-derived NK cells[31]. The de novo-induced CD49a+ NK cells also upregulated Ki67, perforin, granzyme B, and CCL5 (Supplementary Fig. 5b–d) to a greater extent than CD49a− NK cells. The upregulation of

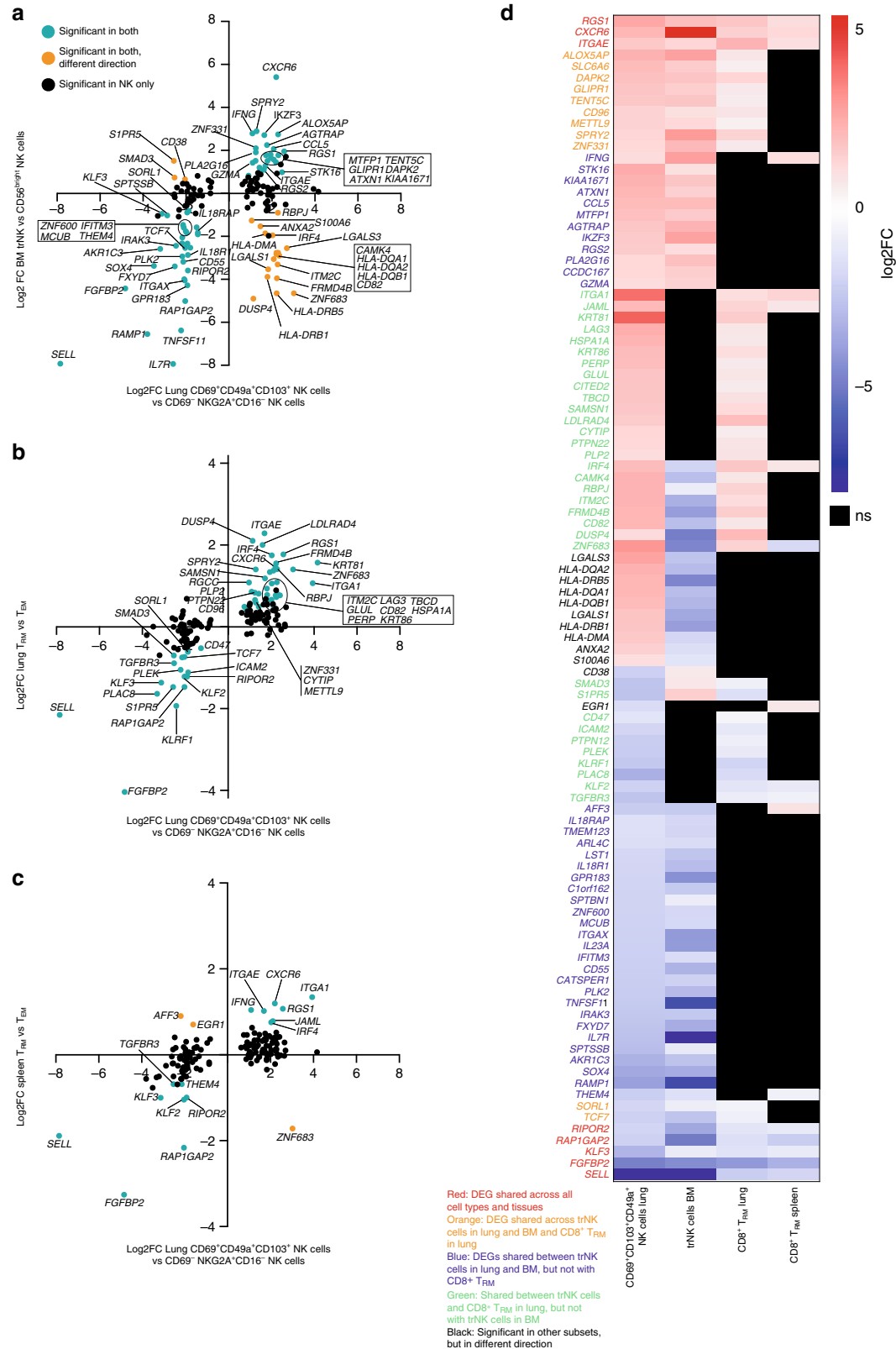

CD49a highlights the need to be cautious when analyzing this response after IL-15 stimulation of bulk NK cells, since de novo CD49a-expressing NK cells could be confounded with bona fide CD49a+ NK cells. Finally, the induction of CD49a on NK cells could indicate that IL-15 plays a role in aiding establishment of tissue-residency in NK cells upon local increase in IL-15 production during viral infections, as has previously been proposed for CD8+ $T_{RM}$ cells[10,32].

Taken together, the present analysis of CD49a+ NK cells, representing the vast majority of trNK cells in human lung,

**Fig. 3** Human lung trNK cells are distinct from trNK cells in bone marrow and CD8[+] T$_{RM}$ cells. Log2 fold-change (log2FC) for differentially expressed genes (padj < 0.05) detected between CD69[+]CD49a[+]CD103[+]NKG2A[+]CD16[−] trNK cells versus CD69[−]NKG2A[+]CD16[−] NK cells in lung plotted against **a** log2FC for trNK cells vs CD56[bright] NK cells in bone marrow, **b** against CD8[+] T$_{RM}$ cells vs T$_{EM}$ cells in lung, and **c** against CD8[+] T$_{RM}$ cells vs T$_{EM}$ cells in spleen. Data for CD8[+] T cells in lung and spleen are from GSE94964[1] and data for NK cells in bone marrow are from GSE116178[20]. In total, 13320 genes detectable above threshold in all datasets were compared. Significantly differentially expressed genes (DEGs) (padj. < 0.05) detected in lung trNK cells only are depicted in black, while DEGs genes shared with the compared subsets are highlighted in green. DEGs significant in both sets but in opposite direction are depicted in orange. **d** Heatmap displaying log2FC of 101 DEGs (padj < 0.05, log2FC >1) in lung CD69[+]CD49a[+]CD103[+]NKG2A[+]CD16[−] NK cells that were also significantly differentially expressed in at least one of the other 3 subsets analyzed (trNK cells in bone marrow and CD8[+] T$_{RM}$ cells in lung and spleen). Gene names in red depicts DEGs shared across all cell types, orange depicts DEGs shared between trNK cells in lung and bone marrow and CD8[+] T$_{RM}$ cells in lung, dark blue depicts DEGs shared between trNK cells in lung and bone marrow, but not with CD8[+] T$_{RM}$ cells in lung and spleen, and green depicts DEGs shared between trNK cells and CD8[+] T$_{RM}$ cells in lung, but not with trNK cells in bone marrow. Gene names in black depicts DEGs significant in trNK cells in bone marrow or in CD8[+] T$_{RM}$ cells, but in opposite direction to trNK cells in lung

reveals that they per se are functional and furthermore are able to upregulate molecules associated with effector functions upon stimulation with IL-15.

## Discussion

There is a growing appreciation that immune cells located in tissues are of key importance in the defense against infections and malignant cells. A better understanding of the nature of tissue-resident lymphocytes, including trNK cells, is therefore not only of importance from a basic scientific standpoint, but also for future development of therapies targeting, or in other ways therapeutically exploiting, these cells. Here, we show that CD16[−] NK cells in the human lung encompass a heterogeneous population of cells. The population contains distinct subsets of NK cells displaying hallmarks of tissue-residency, including expression of CD49a and CD103. Transcriptomic analysis of these subsets revealed that they have a unique transcriptional profile with respect to genes regulating tissue-residency, homing and function. Furthermore, comparative analysis of gene expression-profiles in trNK cells derived from lung and bone marrow, as well as CD8[+] T$_{RM}$ cells from lung and spleen, revealed that gene expression patterns regulating tissue-resident lymphocytes are likely imprinted by the local environments in the respective tissues as well as by cell type.

Our analysis revealed that CD69sp, CD69[+]CD49a[+]CD103[−]CD16[−], and CD69[+]CD49a[+]CD103[+]CD16[−] NK cell subsets had increasingly stronger tissue-resident gene expression signatures, when compared to CD69[−]CD16[−] NK cells. For example, the differences in expression of genes co-regulated across trNK cells and CD8[+] T$_{RM}$ cells in different tissues (Fig. 3d), e.g., *SELL*, *KLF3*, *RGS1*, *RIPOR2*, and *RAP1GAP2*, were gradually increased when comparing CD69spCD16[−], CD49a[+]CD103[−]CD16[−], and CD49a[+]CD103[+]CD16[−] NK cells with CD69[−]CD16[−] NK cells (Fig. S4b). This indicates that there might be different stages of tissue-residency, possibly a reflection of maturation, among trNK cells. Furthermore, the expression patterns of CD69, CD49a and CD103 might also translate to different localization of distinct trNK cell subsets in lung. For example, E-cadherin, the ligand for CD103, and collagen IV, the ligand for CD49a, are expressed by epithelial cells lining the alveolar structures in the lung, and the underlying basement membrane[33], respectively. Human CD103[+] CD8[+] T$_{RM}$ cells have previously been shown to be located within the lung epithelium[14], suggesting that expression of CD103 is indeed indicative of an intraepithelial localization within the lung. In addition, CD69[+]CD103[+]CD8[+] T$_{RM}$ cells are the major CD8[+] T cell subset in epidermis, whereas CD69[+]CD103[−]CD8[+] T$_{RM}$ cells dominate in dermis of human skin[25,34], further supporting the notion that subsets of tissue-resident lymphocytes have distinct localization. It is thus likely that the CD103[+] trNK cells have an intraepithelial location in the lung and are spatially

separated from the CD49a[+]CD103[−]CD16[−] and CD69spCD16[−] NK cells, although the exact localization of the different trNK cell subsets in human lung remains to investigated.

Both CD69[+]CD49a[+]CD103[−] and CD69[+]CD49a[+]CD103[+] trNK cells expressed CXCR3 and CXCR6. CXCR6 has previously been shown to be also expressed by trNK cells in human liver and bone marrow[18,22]. The ligand for CXCR6, the chemokine CXCL16, is produced by alveolar macrophages in high amounts as observed both in health and in different conditions of disease[35]. Thus, CXCR6 might not be unique for homing of NK cells to the liver or bone marrow but might also mediate NK cell homing to the lung. Indeed, expression of CXCR6 was also significantly upregulated by CD8[+] T$_{RM}$ cells in human lung and spleen (Fig. 3d)[1], suggesting a general role of CXCR6 in the homing to several tissues. In addition to CXCR6, several murine models suggest a role for CXCR3 in the migration of T cells and NK cells to the lung under steady-state levels, viral infection, and non-infectious lung-injury[36–38], and antigen-specific CD8[+] T cells in bronchoalveolar lavage (BAL) fluid were homogenously CXCR3[hi]CD49a[+][36]. Although not much is known about the contribution of CXCR3-mediated tissue infiltration by NK cells, CXCR3 is co-expressed on murine CD69[+] NK cells in several tissues[39]. CXCR3 has also been shown to be crucial for the migration of NK cells towards and into solid tumors, which produce high levels of ligands for CXCR3[40,41]. Furthermore, we recently demonstrated a slight but detectable upregulation of CD49a on CXCR3[+] NK cells on human peripheral blood NK cells upon acute infection with influenza A virus[42]. Thus, both CXCR3 and CXCR6 might be involved in the homing of NK cells to the human lung and may, therefore, be important for the generation of a trNK cell population in peripheral tissues (including the liver and lung) as well as in lymphoid tissue (e.g., the bone marrow).

Our comparison of differentially expressed genes with published data from trNK cells in bone marrow and CD8[+] T$_{RM}$ cells in lung and spleen revealed that although trNK cells in lung had a distinct transcriptional profile, they also shared sets of genes with the other tissue-resident NK cells and T cells. The finding that all subsets shared a core of differentially expressed genes, including *SELL*, *S1PR1*, *CXCR6*, *RGS1*, and *KLF3*, indicates that these genes regulate tissue-residency across cell types and tissue locations. Interestingly, the CD69[+]CD49a[+]CD103[+] trNK cell subset in the lung also shared largely distinct sets of genes with trNK cells in bone marrow and with CD8[+] T$_{RM}$ cells in lung, respectively. This indicated that both the environment and cell lineage dictate gene expression patterns in tissue-resident lymphocytes. Furthermore, the expression of a set of genes was remarkably different between trNK cells in lung and bone marrow, including the expression of *ZNF683* and *CD82* (↑ in lung, ↓ in bone marrow), and *S1PR5* and *SMAD3* (↓ in lung, ↑ in bone marrow). Taken together, these

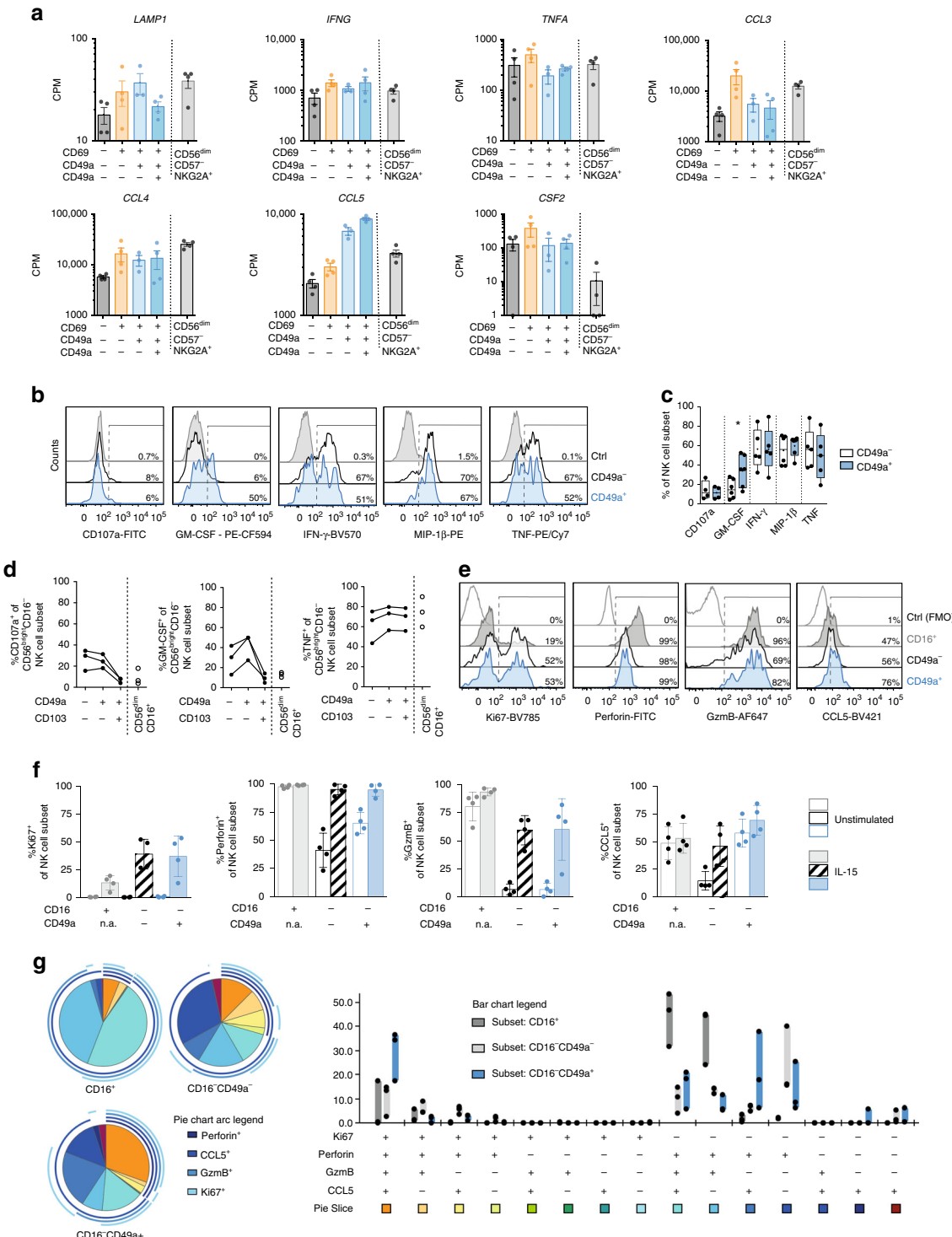

findings further strengthen the notion of the specific tissue location as a factor involved in dictating the transcriptional profile of tissue-resident lymphocytes.

Lung trNK cells were prone to produce CCL5, MIP-1β, and GM-CSF, which might fundamentally alter their local microenvironment by attracting other immune cells such as monocytes, T cells, and eosinophils. Indeed, CCR5, the receptor for MIP-1β and CCL5, is required for efficient CD8[+] T cell recruitment to the lung airways upon viral infection[43], and GM-CSF is critical for terminal differentiation and function of alveolar macrophages[44].

Thus, lung trNK cells might have both pro- and anti-inflammatory influences in respiratory diseases, depending on the effector molecules they produce. Immune activation in the lung during, e.g., viral infection results in elevated levels of IL-15 in the tissue but not in the serum, as demonstrated in influenza infection models[45,46]. IL-15 is a crucial cytokine for lung T cell and NK cell survival, activation, and formation of long-lived $T_{RM}$ cells[10,28,46]. Hence, IL-15 may have a key role also for lung trNK cell activation during viral infections. Indeed, we demonstrate that human lung trNK cells respond strongly towards stimulation

**Fig. 4** Human lung trNK cells are functional upon stimulation. **a** Counts per million (CPM) reads for selected genes in human lung CD69⁻NKG2A⁺CD16⁻ (dark grey), CD69spNKG2A⁺CD16⁻ (orange), CD69⁺CD49a⁺CD103⁻NKG2A⁺CD16⁻(light blue), and CD69⁺CD49a⁺CD103⁺NKG2A⁺ CD16⁻(dark blue) NK cells. CD56^dimCD16⁺CD57⁻NKG2A⁺ NK cells (light grey) are shown for comparison. Two biological replicates from two donors are shown. Mean ± SEM is shown. **b** Representative histograms and **c** summary of data for degranulation and cytokine production of PMA/ionomycin-stimulated lung NK cells. CD49a⁻ and CD49a⁺ NK cells were gated, and background responses in unstimulated controls were subtracted (CD107a, $n = 4$; GM-CSF, $n = 6$; IFN-γ, $n = 5$; MIP-1β, $n = 6$; TNF, $n = 5$). Box and Whiskers, min to max with central line indicating the median and individual data points shown, the mean is indicated as ' + '. Wilcoxon matched-pairs signed rank test, *$p < 0.05$. Numbers in **b** indicate percentage of marker-positive NK cells. **d** Degranulation and cytokine production by sorted CD16⁻ and CD16⁺ human lung NK cells after PMA/ionomycin stimulation. CD49a and CD103 were used to identify distinct NK cell subsets ($n = 3$) after stimulation. **e** Representative overlays and **f** summary of data for expression of Ki67, perforin, granzyme B, and CCL5 in sorted CD16⁺, CD16⁻CD49a⁻, and CD16⁻CD49a⁺ lung NK cells, respectively, following stimulation with IL-15 for 3 days. ($n = 4$). Mean ± SD is shown. n.a. = not analyzed. The majority of the sorted CD16⁻CD49a⁺ NK cells also co-expressed CD103. Numbers in **e** indicate percentage of marker-positive NK cells. **g** SPICE analysis for proliferation and effector molecule expression in sorted CD16⁺, CD16⁻CD49a⁻, and CD16⁻CD49a⁺ lung NK cell subsets following stimulation with IL-15 for 3 days ($n = 3$). Floating bars, min-max. Individual data points are shown. Source data are provided as a Source Data file

with IL-15 and become polyfunctional. Furthermore, de novo IL-15-induced CD49a⁺CD16⁻ lung NK cells were highly responsive, possibly promoting the generation and discrete activation of trNK cells in the lung upon local exposure to IL-15 during, e.g., viral infection.

In summary, our data demonstrate that human lung CD69⁺ CD49a⁺CD103⁻ and CD69⁺CD49a⁺CD103⁺ trNK cells are clearly distinct from other NK cell subsets in the lung and other tissues, whereas CD69spCD16⁻ NK cells (lacking expression of CD49a and/or CD103) are more similar to conventional CD69⁻CD16⁻ NK cells. Furthermore, lung trNK cells are functionally competent and, as such, likely to constitute a first line of defense in the human lung. Together, our characterization of trNK cells in the human lung provides a basis for future studies of these cells in respiratory disease development, progression, and treatment.

## Methods
**Patients**. Tissue samples from a total of 112 patients undergoing lobectomy for suspected lung cancer were collected in the present study. None of the patients received preoperative chemotherapy and/or radiotherapy. Patients with records of strong immunosuppressive medication and/or hematological malignancy were excluded from tissue sampling. The regional review board in Stockholm approved the study, all relevant regulations for work with human participants were complied with, and patient samples were obtained according to the Declaration of Helsinki. All donors gave informed written consent prior to surgery for sample collection. Clinical and demographic details of patients are summarized in Table 1.

**Processing of tissue specimens and peripheral blood**. A small part of macroscopically tumor-free human lung tissue from each patient was transferred into ice-cold Krebs-Henseleit buffer and stored on ice for less than 18 h until further processing. The tissue was digested using collagenase II (0.25 mg/ml, Sigma-Aldrich) and DNase (0.2 mg/ml, Roche), filtered and washed in complete RPMI 1640 medium (Thermo Scientific) supplemented with 10% FCS (Thermo Scientific), 1 mM L-glutamine (Invitrogen), 100 U/ml penicillin, and 50 μg/ml streptomycin (R10 medium). Finally, mononuclear cells from the lung cell suspensions and matched peripheral blood were isolated by density gradient centrifugation (Lymphoprep).

**Flow cytometry**. Antibodies and clones reactive against the following proteins were used (target, clone, fluorochrome, catalog number, dilution): CD3 (UCHT1, PE/Cy5, Beckman Coulter, #A07749, 1/50), CD8 (RPA-T8, Brilliant Violet 570, Biolegend, #301038, 1/50), CD14 (MφP9, Horizon V500, BD Biosciences, #561391, 1/100), CD16 (3G8, Brilliant Violet 711, #302043, 1/50 or Brilliant Violet 785, #302046, 1/200, both Biolegend), CD19 (HIB19, Horizon V500, BD Biosciences, #561121, 1/100), CD45 (HI30, Alexa Fluor 700, Biolegend, #304024, 1/200), CD49a (TS2A, Alexa Fluor 647, Biolegend, #328310, 1/25), CD56 (N901, ECD, Beckman Coulter, #A82943, 1/50, or HCD56, Brilliant Violet 711, Biolegend, #318336. 1/100), CD57 (TB01, purified, eBioscience, #16-0577, 1/100), CD103 (B-Ly7, APC, eBioscience, #17-1038-41, 1/50, or 2G5, biotin, Beckman Coulter, #IM0629, 1/50, or Ber-ACT8, Brilliant Violet 711, BD Biosciences, #563162, 1/50, or Ber-ACT8, PE/Cy7, Biolegend, #350212, 1/50), NKG2A (Z1991.10, APC-A780, Beckman Coulter, custom conjugation, 1/25), CD69 (TP1.55.3, ECD, Beckman Coulter, #6607110, 1/50), CD127 (HIL-7R-M21, Brilliant Violet 421, BD Biosciences,

#562436, 1/50 or R34.34, PE/Cy7, Beckman Coulter, #FA64618, 1/25), CXCR3 (G025H7, PE/Cy7, Biolegend, #353720, 1/25), CXCR6 (K041E5, Brilliant Violet 421, Biolegend, #151109, 1/50), KIR2DL1/S1 (EB6, Beckman Coulter, #A66898, 1/25) and KIR2DL2/3/S2 (GL183, PE/Cy5.5, Beckman Coulter, #A66900, 1/25). After two washes, cells were stained with streptavidin Qdot 605 or Qdot 585 (both Invitrogen), anti-mouse IgM (II/41, eFluor 650NC, eBioscience, #95-5790-42, 1/200) and Live/Dead Aqua (Invitrogen). After surface staining, peripheral blood mononuclear cells (PBMC) were fixed and permeabilized using FoxP3/Transcription Factor staining kit (eBioscience). For intracellular staining the following antibodies were used: Eomes (WF1928, FITC, eBioscience, #11-4877-42, 1/50), CCL5 (2D5, Brilliant Violet 421, BD Biosciences, #564754, 1/25), GM-CSF (BVD2-21C11, PE-CF594, BD Biosciences, #562857, 1/50), granzyme B (GB11, Alexa Fluor 647, BD Biosciences, #560212, 1/50), IFN-γ (4 S.B3, Brilliant Violet 570, Biolegend, #502533, 1/25), Ki67 (B56, FITC, BD Biosciences, #556026, 1/100), MIP-1β (D21-1351, PE, BD Biosciences, #550078, 1/50), perforin (dG9, FITC, BD Biosciences, #556577, 1/50), T-bet (4B10, Brilliant Violet 421, #644816, 1/50 or O4-46, PE-CF594, BD Biosciences, #562467, 1/50), and TNF (MAb11, PE/Cy7, Biolegend, #502929, 1/50). Samples were analyzed on a BD LSR Fortessa equipped with four lasers (BD Biosciences), and data were analyzed using FlowJo version 9.5.2 (Tree Star Inc).

**Table 1 Clinical and demographic details of the 112 patients included in the study**

|  | Non-smoker ($n = 14$) | Current smoker ($n = 29$) | Ex-smoker ($n = 69$) |
|---|---|---|---|
| Female/male | 9 / 5 | 16 / 13 | 44 / 25 |
| Age (year), mean ± SD | 68 ± 12.3 | 66 ± 9 | 68 ± 8.8 |
| FEV1/FVC (% of predicted) mean ± SD | 103 ± 8.8[a] | 91 ± 13.8[b] | 96 ± 14[c] |
| *Pathology* | % (n) | % (n) | % (n) |
| Non-malignant | 7 (1) | 3 (1) | 4 (3) |
| Adenocarcinoma | 50 (7) | 62 (18) | 71 (49) |
| Large cell carcinoma | 0 (0) | 10 (3) | 4 (3) |
| Squamous cell carcinoma | 0 (0) | 10 (3) | 9 (6) |
| Metastasis | 14 (2) | 3 (1) | 4 (3) |
| Carcinoid | 29 (4) | 7 (2) | 6 (4) |
| Other | 0 (0) | 3 (1) | 1 (1) |
| *Medication* | % (n) | % (n) | % (n) |
| Inhaled corticosteroids alone | 14 (2) | 0 (0) | 1 (1) |
| Statins | 43 (6) | 21 (6) | 25 (17) |
| Systemic immunosuppression | 0 (0) | 0 (0) | 4 (3) |
| Beta-agonists or anti-cholinergics | 0 (0) | 7 (2) | 12 (8) |
| Inhaled corticosteroid and long-acting beta-agonist combination | 14 (2) | 7 (2) | 6 (4) |

[a]$n = 13$
[b]$n = 27$
[c]$n = 66$

For cell sorting experiments for RNA sequencing thawed cryopreserved lung mononuclear cells were stained with anti-human CD57 (NK-1, FITC, BD Biosciences, #555619, 1/25), CD16 (3G2, Pacific Blue, BD Biosciences, #558122, 1/50), CD14 (MφP9, Horizon V500, BD Biosciences, #561391, 1/100), CD19 (HIB19, Horizon V500, BD Biosciences, #561121, 1/100), CD103 (Ber-ACT8, Brilliant Violet 711, BD Biosciences, #563162, 1/50), CD49a (TS2/7, AlexaFluor 647, Biolegend, #328310, 1/25), CD45 (HI30, Alexa Fluor 700, Biolegend, #304024, 1/200), CD8 (RPA-T8, APC/Cy7, BD Biosciences, #557834, 1/25), NKG2A (Z199.10, PE, Beckman Coulter, #IM3291U, 1/50), CD69 (TP1.55.3, ECD, Beckman Coulter, #6607110, 1/50), CD3 (UCHT1, PE/Cy5, Beckman Coulter, #A07749, 1/50), KIR2DL1/S1 (EB6, PE/Cy5.5, Beckman Coulter, #A66898, 1/25), KIR2DL2/3/S2 (GL183, PE/Cy5.5, Beckman Coulter, #A66900, 1/25), NKG2C (134591, biotin, R&D Systems, custom conjugate, 1/50), CD56 (NCAM16.1, PE/Cy7, BD Biosciences, #557747, 1/50), streptavidin Qdot655 (Invitrogen), and Live/Dead Aqua (Invitrogen). For cell sorting experiments for functional assays, thawed cryopreserved lung mononuclear cells were stained with CD3 (UCHT1, PE/Cy5, Beckman Coulter, #A07749, 1/50), CD14 (MφP9, Horizon V500, BD Biosciences, #561391, 1/100), CD16 (3G8, APC/Cy7, BD Biosciences, #557758, 1/50), CD19 (HIB19, Horizon V500, BD Biosciences, #561121, 1/100), CD45 (HI30, Alexa Fluor 700, Biolegend, #304024, 1/200), CD49a (TS2/7, PE/Cy7, Biolegend, #328311, 1/50), CD56 (N901, PE/Cy5.5, Beckman Coulter, #A79388, 1/50), CD103 (Ber-ACT8, Brilliant Violet 711, BD Biosciences, #563162, 1/50), and Live/Dead Aqua (Invitrogen). Cells were sorted on a FACSAria Fusion (BD Biosciences).

**Functional analysis of lung NK cells**. In one set of experiments, lung cells were cultured in the absence or presence of phorbol 12-myristate 13−acetate (PMA, 10 ng/ml) and ionomycin (500 nM) for 4 h. Monensin and brefeldin A (GolgiPlug/ Stop, BD Biosciences) were added for the last 3 h of incubation. Anti-human CD107a (FITC, H4A3; BD Biosciences, #555800, 1/100) was present throughout the assay. Finally, cells were stained and analyzed by flow cytometry, as described above. In another set of experiments, sorted lung and blood NK cell subsets were cultured in the presence or absence of IL-15 (10 ng/ml) (Peprotech) for 3 days, followed by staining and analysis by flow cytometry.

**Preparation of libraries for RNA sequencing**. RNAseq was performed using a modified version of the SMART-Seq2 protocol[47]. Four subsets of live, CD3−CD14− CD19−CD56+NKG2A+CD16− NK cells were sorted: CD69−CD49a−CD103− (CD69−), CD69+CD49a−CD103− (CD69sp), CD69+CD49a+CD103−, and CD69+CD49a+CD103+. CD56dimCD16+NKG2A+CD57− NK cells were sorted as a reference. Duplicates of 100 cells from each population from two individual donors were sorted into 4.2 μl of lysis buffer (0.2% Triton X-100, 2.5 μM oligo-dT (5′-AAGCAGTGGTATCAACGCAGAGTACT30VN-3′), 2.5 mM dNTP, RNASe Inhibitor (Takara), and ERCC RNA spike in controls (Ambion)) in a 96-well V-bottom PCR plate (Thermo Fisher). Sorted cells were then frozen and stored at −80 °C until they could be processed. Subsequent steps were performed following the standard SMART-Seq2 protocol with 22 cycles of cDNA amplification and sample quality was determined using a bioanalyzer (Agilent, High Sensitivity DNA chip). Five nanograms of amplified cDNA was taken for tagmentation using a customized in-house protocol[48] and Nextera XT primers. Pooled samples were sequenced on a HiSeq2500 on high output mode with paired 2 × 125 bp reads.

**Transcriptome analysis**. Following sequencing and demultiplexing, read pairs were trimmed from Illumina adapters using cutadapt (version 1.14)[49], and UrQt was used to trim all bases with a phred quality score below 20[50]. Read pairs were subsequently aligned to the protein coding sequences of the human transcriptome (gencode.v26.pc_transcripts.fa) using Salmon (version 0.8.2)[51], and gene annotation using gencode.v26.annotation.gtf. DeSeq2[52] was used to analyze RNA-seq data in R studio version 1.20. Briefly, averaged raw count values for biological replicates were used as input into deSeq2, and variance stabilizing transformation was used to transform data. Data were batch- and patient corrected using Limma[53]. A cut-off of > 100 counts for the four subsets among the 15 samples was used to filter out low expressed genes. Genes with an adjusted $p$-value < 0.05 and a log2-fold change greater than 1 were considered as differentially expressed between samples. Similarly, previously published data sets on trNK cells in bone marrow (GSE116178)[20] and on CD8+ T$_{RM}$ cells in lung and spleen (GSE94964)[1] were analyzed using deSeq2 to identify differentially expressed genes. Heatmaps of gene expression were generated using Pheatmap in R and show z-score for differentially expressed genes (as determined above in deSeq2) for all donors and replicates. Euler diagrams were generated by the eeuler-package in R.

**Statistics**. GraphPad Prism 6 and 7 (GraphPad Software) were used for statistical analysis. For each analysis, measurements were taken from distinct samples. The statistical method used is indicated in each figure legend.

## Data availability
The read counts for the RNA sequencing data derived from lung NK cells is available at GEO with accession number GSE130379. RNA sequencing data for trNK cells in bone marrow was previously published[20] and is available at GEO with accession number GSE116178. RNA sequencing data for CD8+ T$_{RM}$ cells in lung and spleen was previously published[1] and is available at GEO with accession number GSE94964. All other relevant data that support the findings of the study are available from the corresponding author upon reasonable request. The source data underlying Figs. 1b, 1c, 1e–i, 2b, 2d, 2f, 2h, 4a, 4c–d, 4f–g, and Supplementary Figs. S1e, S2a–d, and S5b–d are provided as a Source Data file.

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

## Acknowledgements

We want to thank all donors for participating in the study and P. Bergman, A.-C. Orre, and S. Hylander for administrative and clinical help. We would like to acknowledge the MedH Core Flow Cytometry Facility (Karolinska Institutet), supported by KI/SLL, for providing cell sorting services. The authors acknowledge support from the National Genomics Infrastructure in Stockholm funded by Science for Life Laboratory, the Knut and Alice Wallenberg Foundation and the Swedish Research Council, and SNIC/Uppsala Multidisciplinary Center for Advanced Computational Science for assistance with massively parallel sequencing and access to the UPPMAX computational infrastructure. This work was supported by the Swedish Research Council, the Swedish Foundation for Strategic Research, the Foundation for Strategic Research, the Swedish Cancer Society, and the Eva and Oscar Ahrén Research Foundation. J.M. received funding from the Swedish Heart-Lung Foundation, M.L. received funding from the Swedish Children's Cancer Foundation. Open access funding provided by Karolinska Institute.

## Author contributions

Conceptualization: N.M., H.G.L., J.M.; Methodology: N.M., E.K., J.H., J.E.M., J.M.; Investigation: N.M., E.K., P.C., M.L., J.N.W., M.S., J.M.; Resources: P.B., M.A-A.; Writing – original draft: N.M., J.M.; Writing–review and editing: N.M., J.M., H.G.L; Visualization: N.M., J.M.; Funding acquisition: N.M., J.M., H.G.L.

## Additional information

**Competing interests:** The authors declare no competing interests.

