## [Peer Review File · Nature Communications]

Reviewers' comments:

Reviewer #1 (Remarks to the Author):

The authors address interesting questions in the field of tissue-resident NK cells and compare lung and bone marrow trNK cells, and CD8 Trm cells from spleen and lung. The RNAseq data on trNK cells in the lung are novel and these data are compared with published data. Data on gene expression, flow cytometry and stimulation experiments are provided. The authors claim the presence of sets of core genes associated with tissue-residency or location.

A major problem is the readability of the manuscript. The definition of the NK cell subsets is not consistent. TrNK cells are defined in the text by the expression of CD69 and 'co-expressing' CD49a and/or CD103 line 27. This formulation is not unequivocal, and creates confusion reading the rest of the manuscript.

Major points

- The full gating strategy of NK cells in lung should be provided including relative composition. It is likely that lung may include the CD56bright and CD56dim subsets present in blood, plus extra CD69+ population(s). Of those CD69+ cells certain fraction will express CD49a and/or CD103. Such an overview with relative contributions to the whole NK population would provide a start of the manuscript to help better understand the distribution of the various subpopulations. The spread in distribution among individuals could maybe also explain the differences between the density plot in fig S1 and fig 1a?

- The authors suggest based on the PCA analysis in figure 2a (performed only on the 1000 most variable genes) and differentially expressed genes (fig 2b), that CD69SP cells are most similar to CD69-CD56bright cells. However, in fig2d, CXCR6, EOMES, IFNG are higher expressed while SELL and S1PR1 are lower expressed in CD69SP compared to CD69-CD56bright (probably because of the low significance (as a consequence of n=2, the aforementioned genes are not included in the list of DEG). This gene signature is characteristic for tissue-resident NK cells in both liver and bone marrow. It's very interesting that the authors investigated the differences and similarities between BM and lung trNK cells. But, what would be the result if the CD69sp NK cells in lung are compared with the CD69+ BM trNK cells. The author should discuss this in their manuscript and add a comparison of the CD69SP cells with the trNK cells in bone marrow.

- Most figure legends identify trNK cells as CD49a/CD103+, but which cells are meant? The definition "CD49a+ and/or CD103+" used throughout the text of the paper, suggests UL, UR, and LR quadrant of fig1a, but RNAseq data are only provided on cells in the UL and UR quadrant. From Figure 4B onwards, cells are defined as CD49a+ or - .

- Then the authors point out that trNK cells (CD49a+ and/or CD103+) are mainly CD56brightCD16neg, and RNAseq and protein analyses are performed without inclusion of the CD56dim population (although CD56dimCD16+ cells were sorted as reference, as indicated in M&M). Why are the CD56 dim cells not included in the gene expression comparisons? The authors should include the RNA sequence data of the CD56dim NK cells in their manuscript as well to fairly compare all the different NK cell subsets (with CD8+ T cells).

- The subset definition of the NK cells and the colors used to indicate the subsets differ between the figures. The authors should use throughout the article the same definition of each subset for both the RNA expression data and the protein data.

- The authors state that they would like to determine differences between lung trNK cells and peripheral blood NK cells, however no data is included on peripheral blood NK cells. If available, they should include these data.

- Technical duplicates (100 cells !!!) of each donor sample were sequenced. 100 cells per sample is very few cells and this may limit the significance of the data. The results of the technical duplicates, which are not independent, are represented in the figures as being biological. It would be more fair to pool the data of the replicates.

- The statement at the end of the discussion that trNK cells constitute a first line of defense due to their intraepithelial localization and functional capacity to respond to cytokine stimulation, is very

attractive, but intraepithelial localization is not shown.

- As for the responsiveness, from figure 5 it can be concluded that IL-15 upregulates CD49a and the cells that are well stimulated and become CD49a pos after stimulation are also most stimulated to express Ki67, GZMB and CCL5. If the aim was to generate trNK cells from peripheral blood NK cells it should be clarified. Are these cells then similar to trNK cells?

Figure 5 in general requires more attention in the text and needs to be linked to the previous findings.

Minor points

- Abstract mentions spleen trNK cells?

- In figure 1c the text states that almost all lung NK cells expressing CD49a and/or CD103 express CD69. However, the figure is represented in the opposite way: the majority of CD69+ NK cells expresses CD49a (there is still a huge variation between patients, which should be noted). Moreover, the CD103 expression is lacking in Figure 1c.

- CD49a and CD103 are not expressed on the ILCs, so therefore they are not intraepithelial? The statements about CD103 and intraepithelial localization should be supported by data.

- Some explanation is required on the term trNK cells in the RNA sequencing analysis. For instance: since the CD69+CD49a+CD103-, CD69+CD49a+CD103+ populations did not notably differ, we pooled these samples in further analyses (termed trNK).

- The zero on the heatmap scales should be positioned in the white area not in the yellow area. Now its misleading: yellow means equal to the mean, rather than upregulated.

- Figure 1a,c: the legend states CD69-, CD69-CD56bright would be more precise.

- Figure 2c does not show which genes are significantly differently expressed, without adding the gene names this figure is not of any additional value (the number of DEGs can also be deduced from the Venn diagram).

- Figure 2d: which genes are differentially expressed? There is no information on significance included.

- Figure 2g: The mean CD16 expression is much higher than shown in the 'representative' histogram in figure 2e, is this correct?

- The CXCR6 protein expression in figure 2f seems to be only a matter of intensity difference so its not fair to set a gate and express CXCR6 expression as % positive. Could it be that the collagenase treatment is influencing your CXCR6 expression? Moreover, line 162 states that trNK cells express CXCR6 but looking at the figure only 20% is positive.

- The CD69- CD56bright NK cells seem to have a high CD57 expression (Figure 2h), this is unlikely, could this be caused by contamination of CD56dim NK cells?

- The legends lack information whether the bargraphs show mean/median SD or SEM.

- Line 142: the references to the figures are lacking. There is no statement on statistics in either figure or text.

- The section on the functional data states that CCL5 and CSF1 are highly expressed in trNK cells, but for CSF1 this is only high in comparison with other NK cells, so this statement should be put in perspective.

- In the discussion it is stated '...assessed phenotype, transcriptome and function in 104 lungs at an unprecedented level', but except for fig 1, I do not see so many data points in the figures. This is an overstatement.

- Data obtained from smokers are not really contributing to the message of the paper, except that the frequency of trNK cells is a bit increased in some individuals.

Reviewer #2 (Remarks to the Author):

The manuscript entitled: Unique transcriptional and protein-expression signature in human lung tissue resident NK cells by Nicole Marquardt et al. describes NK cell genetic and phenotypic landscape in human lung using RNA sequencing and 18colour flow cytometry compared to lung, bone marrow and spleen NK cells and CD8 Trm cells. Core data sets are revealed and validated that are co-regulated by

tissue residency, cell type or location. The beauty of the study is that it is conducted with human tissues (eg lung) that are difficult to obtain and that a high number of samples is investigated. The paper is well conducted written. The paper would be strengthened by a clear take home message with regards to NK cells relevant to disease.

In mouse experiments, tissue residency can be experimentally addressed by parabiosis experiments. In human there is no experimental possibility and thus the word "tissue residency" should be used with more caution throughout the paper. I find it confusing to call CD69+CD49a+ and/or CD103+ "trNK" cells (as done in figure 2b).

Major points:

- In figure 1, NK numbers of smokers and non smokers are shown. Are there additional data available about the phenotype changes in these cohorts? A more comprehensive analysis of phenotypical and functional changes in lung NK cells during disease would strengthen the paper. The authors describe in the result section that frequencies of NK cells were not affected by the medication taken by the patients (statins and corticoids. The authors should also address and provide data whether the major phenotypic changes reported in lung were affected by the medication taken by the patients.
- Were differences in NK Phenotypes observed depending on the cancer types? Was a correlation to prognosis detectable?
- Fig 2: CD49a/CD103: does this mean and/or (as commented in the figure legend) or both molecules to be co-expressed? Please clarify
- Fig 3: CD8+ TRM cells vs TEM how were these cells isolated? This should be included in the result section
- Fig 4: Polyfunctional NK cells: Do the same NK cells kill and produce cytokines? Are these different subsets? Is TRAIL expressed? How does the Digestion of the tissue Change different marker Expression?
- Figure 5: the in vitro stimulation with IL-15 does not add much novel information (compared to ref 27) and appears out of context.

Minor comments

1. Abstract: polyfunctional: please spell out the functional characteristics, this could help to clearly state novelty of the study and the take home message
2. Abstract: last sentence: unique part..unclear what the authors mean with this
3. Page 6, 108: non-NK ILC include how defined in text,
4. Figure 2. It is explained in the result section that protein expression of Eomes is highest in trNK cells. However, it seems that mean value is same as in CD69sp NK cells.
5. Figure 4. clarify in the figure legend what n.a. means for CD49a on the graphs f and e.

Point-by-point response to the reviewers' comments:

Reviewer #1 (Remarks to the Author):

The authors address interesting questions in the field of tissue-resident NK cells and compare lung and bone marrow trNK cells, and CD8 Trm cells from spleen and lung. The RNAseq data on trNK cells in the lung are novel and these data are compared with published data. Data on gene expression, flow cytometry and stimulation experiments are provided. The authors claim the presence of sets of core genes associated with tissue-residency or location. A major problem is the readability of the manuscript. The definition of the NK cell subsets is not consistent. TrNK cells are defined in the text by the expression of CD69 and 'co-expressing' CD49a and/or CD103 line 27. This formulation is not unequivocal, and creates confusion reading the rest of the manuscript.

We appreciate the careful review of the manuscript provided by the reviewer and have now substantially revised the manuscript in line with the comments and constructive criticism provided. In particular, we have made an effort to enhance the clarity of the paper, including more well-defined definitions of the different NK cell subsets analysed throughout the manuscript. Below, we provide a point-by-point response to the comments made by the reviewer.

Major points

- The full gating strategy of NK cells in lung should be provided including relative composition. It is likely that lung may include the CD56^{bright} and CD56^{dim} subsets present in blood, plus extra CD69⁺ population(s). Of those CD69⁺ cells certain fraction will express CD49a and/or CD103. Such an overview with relative contributions to the whole NK population would provide a start of the manuscript to help better understand the distribution of the various subpopulations. The spread in distribution among individuals could maybe also explain the differences between the density plot in fig S1 and fig 1a?

Answer:

In the revised manuscript we provide a full gating strategy (Fig. S1a). To clarify the overall contribution of different NK cell subsets to the total NK cell population in lung, we now provide results showing the frequency of 16 distinct NK cell subsets based on the expression of CD16, CD69, CD49a, and CD103 out of total NK cells (Fig. 1g in revised manuscript). The new data show that CD16⁻ NK cells expressing different combinations of CD49a and CD103 make up a substantial proportion of the total NK cell population ($\approx 20\%$ of total NK cells), although as expected from our previous report (Marquardt *et al.*, 2017), CD56⁺CD16⁺CD69⁻ is clearly the major population of NK cells detected in human lung. CD49a and CD103 are not expressed at detectable levels on blood NK cells (Fig. S1b in revised manuscript), and as such expression of these proteins clearly distinguishes discrete NK cell subsets present in lung, but not in blood.

In addition to the overall distribution of NK cell subsets, we also provide the distribution of NK cells expressing CD49a and/or CD103 among CD16⁻CD69⁺ and CD16⁺CD69⁺ NK cells (Fig. 1a and b in revised manuscript), which clearly shows that CD49a and CD103 expression is mainly restricted to CD16⁻CD69⁺ NK cells. Interestingly, however, the comparison of CD16⁻CD69⁺ and CD16⁺CD69⁺ NK cells revealed relatively high expression of CD49a and CD103 on CD16⁺CD69⁺ NK cells in a limited number of the donors. In these donors, we identified a small CD56^{bright}CD16⁺ NK cell subset which displayed increased expression of

CD49a and CD103 (Fig. S1d and e), accounting for the spread within the CD16⁺CD69⁺ NK cell subset noted by the reviewer.

- The authors suggest based on the PCA analysis in figure 2a (performed only on the 1000 most variable genes) and differentially expressed genes (fig 2b), that CD69SP cells are most similar to CD69-CD56^{bright} cells. However, in fig2d, CXCR6, EOMES, IFNG are higher expressed while SELL and SIPR1 are lower expressed in CD69SP compared to CD69-CD56^{bright} (probably because of the low significance (as a consequence of n=2, the aforementioned genes are not included in the list of DEG). This gene signature is characteristic for tissue-resident NK cells in both liver and bone marrow. It's very interesting that the authors investigated the differences and similarities between BM and lung trNK cells. But, what would be the result if the CD69sp NK cells in lung are compared with the CD69+ BM trNK cells. The author should discuss this in their manuscript and add a comparison of the CD69SP cells with the trNK cells in bone marrow.

Answer:

The reviewer points out that a few differentially expressed genes between CD69sp and CD69⁻CD56^{bright} NK cells in the lung fit with previously described characteristics of trNK cells in bone marrow and liver. Indeed, out of the relatively few detected differentially expressed genes in CD69sp NK cells in lung (~20 genes), the majority were also differentially expressed in bone marrow when performing a direct comparison of differentially expressed genes between available datasets (Fig. S3a in revised manuscript). It should however be noted that substantially more genes were differentially expressed between CD103⁺CD49a⁺ or CD103⁻CD49a⁺ NK cells and CD69⁻CD56^{bright}CD16⁻ NK cells (~150 genes) compared to comparisons between CD69sp NK cells and CD69⁻CD56^{bright}CD16⁻ NK cells, respectively (Fig. S2b in revised manuscript). Given that we had similar power to detect differentially expressed genes between these comparisons, it is clear that CD69sp are more similar to CD69⁻CD56^{bright}CD16⁻ NK cells in the lung compared to CD103⁺CD49a⁺ and CD103⁻CD49a⁺ NK cells. That said, we agree with the reviewer that CD69sp NK cells do share some traits with trNK cells, both in lung and bone marrow, which we acknowledge and discuss in the revised manuscript (page 9, lines 174-192; page 16, line 346).

- Most figure legends identify trNK cells as CD49a/CD103+, but which cells are meant? The definition "CD49a+ and/or CD103+" used throughout the text of the paper, suggests UL, UR, and LR quadrant of fig1a, but RNAseq data are only provided on cells in the UL and UR quadrant. From Figure 4B onwards, cells are defined as CD49a+ or - .

Answer:

In the revised manuscript, we have clarified the definitions of different NK cell subsets throughout the entire manuscript. For protein analyses, we have applied the Boolean gate "CD69+ AND (CD49a+ OR CD103+)", thus including CD69⁺CD49a⁺CD103⁺, CD69⁺CD49a⁺CD103⁻, and CD69⁺CD49a⁻CD103⁺ NK cells, which we together denote as "CD49a⁺ and/or CD103⁺ NK cells". In cases where this definition cannot be used, e.g., following *in vitro* stimulation where CD69 is upregulated, we have clearly stated which subset of NK cells we refer to.

We agree that it would have been interesting to add RNAseq data from CD69⁺CD49a⁻CD103⁺ NK cells to the analysis. However, the very low frequency of these NK cells in most donors prevented collection of a sufficient number for performing RNAseq.

Given that CD69⁺CD49a⁺CD103⁺ and CD69⁺CD49a⁺CD103⁻ CD16⁻ NK cells were very similar based on RNA expression, with only a handful of differentially expressed genes between them, we decided to combine these subsets to one for most subsequent analysis. As suggested by the reviewer (see comment below) we also explicitly provide this as the rationale for analysing these cells as one population in the revised manuscript (page 10, lines 208-213).

- Then the authors point out that trNK cells (CD49a⁺ and/or CD103⁺) are mainly CD56^{bright}CD16^{neg}, and RNAseq and protein analyses are performed without inclusion of the CD56^{dim} population (although CD56^{dim}CD16⁺ cells were sorted as reference, as indicated in M&M). Why are the CD56^{dim} cells not included in the gene expression comparisons? The authors should include the RNA sequence data of the CD56^{dim} NK cells in their manuscript as well to fairly compare all the different NK cell subsets (with CD8⁺ T cells).

Answer:

The main focus of the study was to analyse the distinct properties of potential trNK cells (here defined as CD16⁻CD69⁺CD49a⁺ and/or CD103⁺ NK cells) in lung, and we therefore compared them to the most phenotypically similar NK cell subsets, *i.e.*, CD69⁻CD16⁻CD56^{bright} NK cells in lung. Comparing trNK cells with CD56^{dim} NK cells would instead most likely reveal general differences between CD56^{dim} and CD56^{bright} NK cells which are not specific for trNK cells. In addition, several previous studies have performed comparative analysis of CD56^{bright} versus CD56^{dim} NK cells (see, *e.g.*, ref. 20-22 in the manuscript). Hence, we felt that a more detailed comparison was out of the scope of the current manuscript. Nevertheless, we do provide a principal component analysis of CD56^{dim}CD16⁺NKG2A⁺CD57⁻ NK cells and CD69⁺CD49a⁺CD103⁺, CD69⁺CD49a⁺CD103⁻, CD69^{sp}, and CD69⁻ CD56^{bright}CD16⁻ NK cells, showing that CD56^{dim}CD16⁺NKG2A⁺CD57⁻ NK cells are clearly distinct from all the other subsets, in line with previous findings (ref. 20-22). The new data are displayed in Fig. S2a. In addition, we provide the raw counts from the RNAseq analysis of CD56^{dim}CD16⁺NKG2A⁺CD57⁺ NK cells as a supplementary dataset, enabling the reader to do a direct comparison of these cells with the subsets of CD16⁻CD56^{bright} NK cells.

- The subset definition of the NK cells and the colors used to indicate the subsets differ between the figures. The authors should use throughout the article the same definition of each subset for both the RNA expression data and the protein data.

Answer:

The colours in the graphs have now been adjusted. In detail, in the analyses including discrimination of CD69^{sp} NK cells (protein and RNA), CD69⁻CD49a⁻CD103⁻ NK cells are depicted as black/dark grey, CD16⁻CD69^{sp} NK cells as orange, and CD16⁻CD69⁺ NK cells co-expressing CD49a and/or CD103 as blue.

However, this gating could not be used entirely throughout the manuscript, *e.g.*, due to upregulation of CD69 in functional assays and some limitations in data acquisition. In analyses comparing CD49a⁻ vs CD49a⁺ NK cells irrespective of CD69 or CD103 expression, CD49a⁻ NK cells are depicted as white with black border, and NK cells expressing at least one of the markers CD49a or CD103 in blue.

By doing like this, we highlight NK cells with a tissue-resident phenotype indicated by expression of CD49a or CD103 throughout the manuscript, facilitating the comparison of the data.

- The authors state that they would like to determine differences between lung trNK cells and peripheral blood NK cells, however no data is included on peripheral blood NK cells. If available, they should include these data.

Answer:

We agree that some phrasing in the discussion (originally line 288) was misleading. The study assessed the characteristics of trNK cells on a phenotypic and transcript level with a particular focus on the human lung. We have revised the text accordingly and removed data from peripheral blood where required, e.g. the original Fig. 5 (Fig. S4 in revised manuscript).

- Technical duplicates (100 cells !!!) of each donor sample were sequenced. 100 cells per sample is very few cells and this may limit the significance of the data. The results of the technical duplicates, which are not independent, are represented in the figures as being biological. It would be more fair to pool the data of the replicates.

Answer:

In the revised manuscript we have, as suggested by the reviewer, performed the analysis on averaged RNAseq data for the replicate biological samples, with largely similar results. Given that we use 100 cells per biological replicate, there is indeed some variation between replicates. In the revised manuscript, we therefore displayed the expression levels for each replicate for differentially expressed genes detected from the averaged data in order for the reader to be able to assess the variation between the biological replicates (Fig. 2c and Fig. S2b in the revised manuscript).

- The statement at the end of the discussion that trNK cells constitute a first line of defense due to their intraepithelial localization and functional capacity to respond to cytokine stimulation, is very attractive, but intraepithelial localization is not shown.

Answer:

We agree with the reviewer that showing that IL-15 results in upregulation of CD49a on NK cells is not entirely novel, since it was previously demonstrated for NK cells in peripheral blood (Hydes et al., ref. 31 in the manuscript). However, we show here for the first time that these *de novo* CD49a⁺ lung NK cells display higher activation levels compared to CD49a⁻ NK cells when stimulating sorted CD16⁺ and CD16⁻CD49a⁻ NK cells with IL-15. The aim of the analysis, however, was primarily made to highlight the fact that these newly emerged CD49a⁺ NK cells might confound results when analysing total CD49a⁺ NK cells after IL-15 stimulation, unless defined populations of sorted CD49a⁺ and CD49a⁻ NK cells are used for the assays (as was done in the current manuscript). That said, the IL-15-induced CD49a expression might aid NK cells in establishing tissue-residency, e.g. upon viral infection in the lung which induces local IL-15 production, as has previously been suggested for tissue-resident T cells. We do however understand the reviewer's opinion that the data appear out of context. Therefore, we moved former Fig. 5 to the supplements (new Fig. S4), slightly revised the figure and the text, and added a clear justification in the respective results section of the manuscript.

- As for the responsiveness, from figure 5 it can be concluded that IL-15 upregulates CD49a and the cells that are well stimulated and become CD49a pos after stimulation are also most stimulated to express Ki67, GZMB and CCL5. If the aim was to generate trNK cells from peripheral blood NK cells it should be clarified. Are these cells then similar to trNK cells? Figure 5 in general requires more attention in the text and needs to be linked to the previous findings.

Answer:

The reviewer is correct in his/her conclusion that IL-15 can upregulate CD49a on CD49a⁻ NK cells in the lung, as has been demonstrated previously for NK cells in peripheral blood (Hydes et al., ref. 31 in the manuscript). However, we show here for the first time that these *de novo* CD49a⁺ lung NK cells display higher activation levels compared to CD49a⁻ NK cells when stimulating sorted CD16⁺ and CD16⁻CD49a⁻ NK cells with IL-15. The aim of the analysis, however, was not to address *per se* whether *de novo*-generated CD49a⁺ NK cells from blood or lung share traits with *bona fide* trNK cells. Rather, we wanted to highlight the fact that upregulation of CD49a during IL-15 stimulation with subsequent analysis of CD49a⁺ NK cells might confound the results, unless defined populations of sorted CD49a⁺ and CD49a⁻ NK cells are used for the assays (as was done in the current manuscript). However, the IL-15-induced CD49a expression might aid NK cells in establishing tissue-residency, e.g., upon viral infection in the lung which induces local IL-15 production, as has previously been shown for tissue-resident T cells.

In response to a further comment from reviewer #2 and in order to avoid any confusion in the relevance of the data in Fig. 5, we have now moved results commented upon above to the supplements (new Fig. S4) and also removed results from IL-15 stimulation of peripheral blood NK cells from both Fig. 4 and Fig. 5. Accordingly, we have revised the text and figure legend as requested by the reviewer.

Minor points

- Abstract mentions spleen trNK cells?

Answer:

We agree that the information in the abstract was misleading. We have now revised the abstract, clarifying that trNK cell datasets were available from the lung and from bone marrow, whereas datasets of CD8⁺ T_{RM} cells from lung and spleen were used (page 2, line 29/30).

- In figure 1c the text states that almost all lung NK cells expressing CD49a and/or CD103 express CD69. However, the figure is represented in the opposite way: the majority of CD69⁺ NK cells expresses CD49a (there is still a huge variation between patients, which should be noted). Moreover, the CD103 expression is lacking in Figure 1c.

Answer:

We now provide analysis matching the reviewer's statement (new Fig. 1f). In line with another comment of reviewer #1, we have replaced the original Fig. 1c with data showing the overall contribution of NK cells expressing CD69, CD49a, and/or CD103 within the NK cell population in human lung (new Fig. 1g). Furthermore, we provide a more detailed analysis showing the distribution of CD49a⁺ and/or CD103⁺ NK cells within CD16⁻CD69⁺ and CD16⁺CD69⁺ NK cells, respectively (Fig. 1b). The text in the results has been revised accordingly (page 6, lines 105-107; lines 111-116).

- *CD49a and CD103 are not expressed on the ILCs, so therefore they are not intraepithelial? The statements about CD103 and intraepithelial localization should be supported by data.*

Answer:

We agree that lack of expression of CD103 and CD49a is rather inconclusive regarding the localization of ILCs in the lung. In response to this, and other comments about the localization of NK cells and non-NK ILCs based on expression of CD103 without providing substantial supporting data, we have now toned down the conclusion in the manuscript (page 7).

- *Some explanation is required on the term trNK cells in the RNA sequencing analysis. For instance: since the CD69⁺CD49a⁺CD103⁻, CD69⁺CD49a⁺CD103⁺ populations did not notably differ, we pooled these samples in further analyses (termed trNK).*

Answer:

When analyzing CD69⁺CD49a⁺CD103⁺ and CD69⁺CD49a⁺CD103⁻ CD16⁻ NK cells separately at the RNA level, only few differentially expressed genes could be identified. Hence, as recommended by the reviewer, we decided to combine these subsets to one for subsequent analyses (where not else noted) (page 10, lines 208-213).

In response to this, and a similar comment by reviewer #1, we have now clarified the definition of the different NK cell subsets used in the RNA sequencing analysis.

- *The zero on the heatmap scales should be positioned in the white area not in the yellow area. Now its misleading: yellow means equal to the mean, rather than upregulated.*

Answer:

In the revised manuscript, we have changed the heatmap scales to blue-white-red (which was previously blue-yellow-red).

- *Figure 1a,c: the legend states CD69⁻, CD69⁻CD56^{bright} would be more precise.*

Answer:

In fact, only CD69⁻CD16⁻ NK cells have been analysed, not CD56^{bright}CD16⁻ NK cells (except for RNAseq analyses and for Fig. S2e). We admit that this was formulated in a misleading way in the results section. We have revised the text in the results accordingly, and applied this to newly added graphs in Figure 1.

- *Figure 2c does not show which genes are significantly differently expressed, without adding the gene names this figure is not of any additional value (the number of DEGs can also be deduced from the Venn diagram).*

Answer:

In the revised manuscript, we now show gene names displaying differentially expressed genes in the heat map.

- Figure 2d: which genes are differentially expressed? There is no information on significance included.

Answer:

In the revised manuscript, we have removed the additional heatmaps (previous Fig 2d), and instead highlight genes in the larger heatmap containing significant differentially expressed genes (Fig. 2c and Fig. S2b in revised manuscript). Additionally, we have added information on statistical significance in the figure legend (page 31, line 697).

- Figure 2g: The mean CD16 expression is much higher than shown in the 'representative' histogram in figure 2e, is this correct?

Answer:

We agree that the displayed histogram is not clearly representing the summarized data in the graph. Thus, we have replaced the histogram with a more representative example (new Fig. 1d and e, included in the manuscript).

- The CXCR6 protein expression in figure 2f seems to be only a matter of intensity difference so its not fair to set a gate and express CXCR6 expression as % positive. Could it be that the collagenase treatment is influencing your CXCR6 expression? Moreover, line 162 states that trNK cells express CXCR6 but looking at the figure only 20% is positive.

Answer:

We agree with the reviewer that it is fairer to analyse the expression levels of CXCR6 rather than the frequency of positive cells. We have changed the corresponding figure accordingly (new Fig. 2e, included in the manuscript).

Furthermore, it is a fair assumption of the reviewer that the collagenase treatment is influencing CXCR6 expression. We have now analysed CXCR6 expression levels on lung trNK cells that have been isolated with or without enzyme treatment from three donors (Fig. 1a below, not included in the manuscript) and found that CXCR6 MFI is slightly decreased after enzyme digestion. Finally, in response to a comment from the second reviewer, we compared the expression of diverse markers including CXCR6 on tonsil cells that were either untreated or enzyme-treated (Fig. 1b below, not included in the manuscript). Similar as in the lung, CXCR6 expression levels were decreased on enzyme-treated tonsil NK cells. However, it is reasonable to assume that the enzyme digestion affected all NK cell subsets equally in the lung tissue. Hence, while overall expression levels might be slightly altered, the observed differences between NK cell subsets observed are likely to be independent from the enzyme treatment.

Fig. 1. Enzyme-treatment reduces expression of CXCR6 on tissue NK cells. (a) Expression levels of CXCR6 on trNK cells (CD56^{bright}CD16⁺CD69⁺ NK cells co-expressing CD49a and/or CD103) from fresh human lungs that were or were not

enzymatically treated are depicted. (b) Expression levels of CXCR6 on total NK cells from cryopreserved mononuclear cells from one human tonsil that was or was not enzymatically treated are shown.

- *The CD69⁻ CD56^{bright} NK cells seem to have a high CD57 expression (Figure 2h), this is unlikely, could this be caused by contamination of CD56^{dim} NK cells?*

Answer:

We would like to emphasize that the respective analysis has been performed on total CD16⁻ NK cells only, and not on CD56^{bright}CD16⁻ NK cells, as misleadingly stated in the manuscript. We have changed the text accordingly. Nevertheless, we agree with the reviewer that the CD16⁻ NK cell population might contain a small subset of CD56^{dim} NK cells that have lost CD16.

- *The legends lack information whether the bargraphs show mean/median SD or SEM.*

Answer:

The bar graphs show mean \pm SD where depicted, except for Fig. 4a, where mean \pm SEM is shown. This information has now been added to the figure legends.

- *Line 142: the references to the figures are lacking. There is no statement on statistics in either figure or text.*

Answer:

The respective results part is related to the original Fig. 2d. The respective heatmaps have been deleted from the figure. Nevertheless, we have now added information about statistics to the text and figure legends for each figure where required.

- *The section on the functional data states that CCL5 and CSF1 are highly expressed in trNK cells, but for CSF1 this is only high in comparison with other NK cells, so this statement should be put in perspective.*

Answer:

The reviewer is correct in stating that we only compare within the lung NK cell population. The CSF1 RNA levels were indeed low, and in the new analyses using averaged data from replicates CSF-1 was no longer a significantly differentially expressed gene. We have therefore removed CSF1 data from Figure 4 in the revised manuscript.

- *In the discussion it is stated ‘...assessed phenotype, transcriptome and function in 104 lungs at an unprecedented level’, but except for fig 1, I do not see so many data points in the figures. This is an overstatement.*

Answer:

We agree with the reviewer that mentioning the total number of donors included in this study is misleading in this context. Therefore, we have now omitted this information.

- Data obtained from smokers are not really contributing to the message of the paper, except that the frequency of trNK cells is a bit increased in some individuals.

Answer:

We agree with the reviewer that the data about the frequency of trNK cells in correlation to the smoking status is not directly relevant for the message of the paper. We have now decided to move the respective data into the new Fig. S1f and have changed the reference in the text accordingly (page 7, line 122).

Reviewer #2 (Remarks to the Author):

The manuscript entitled: Unique transcriptional and protein-expression signature in human lung tissue resident NK cells by Nicole Marquardt et al. describes NK cell genetic and phenotypic landscape in human lung using RNA sequencing and 18colour flow cytometry compared to lung, bone marrow and spleen NK cells and CD8 Trm cells. Core data sets are revealed and validated that are co-regulated by tissue residency, cell type or location. The beauty of the study is that it is conducted with human tissues (eg lung) that are difficult to obtain and that a high number of samples is investigated. The paper is well conducted written. The paper would be strengthened by a clear take home message with regards to NK cells relevant to disease.

In mouse experiments, tissue residency can be experimentally addressed by parabiosis experiments. In human there is no experimental possibility and thus the word "tissue residency" should be used with more caution throughout the paper. I find it confusing to call CD69⁺CD49a⁺ and/or CD103⁺ "trNK" cells (as done in figure 2b).

Answer: We agree with the reviewer that it is extremely difficult, if possible at all, to demonstrate tissue-residency of lymphocytes in humans. The term "tissue-resident" has nevertheless been used in a large number of publications describing human lymphocytes with a very similar phenotype as those shown to be truly tissue-resident by, e.g., parabiosis experiments in mice. Indeed, the CD69⁺CD49a⁺ and/or CD103⁺ NK cells we identify in this study do also share a gene- and protein-expression signature with cells that have been shown to be tissue-resident in mice as revealed by parabiosis experiments, including elevated expression of CD103, CD49a, CD69, and *ZNF683*, and lower expression of *SELL*, *SIPR1*, and *SIPR5*. In the revised manuscript we are more cautious using the term "tissue-resident" and acknowledge that fact that although sharing key signatures with mouse tissue-resident cells, we cannot formally prove that these cells are truly tissue-resident.

Major points:

• *In figure 1, NK numbers of smokers and non-smokers are shown. Are there additional data available about the phenotype changes in these cohorts? A more comprehensive analysis of phenotypical and functional changes in lung NK cells during disease would strengthen the paper.*

Answer:

We have previously (Marquardt et al., JACI 2016) shown that both the NK cell frequency and degranulation in response to target cells are altered in relation to the cigarette smoking status. In detail, NK cell frequencies in blood and lung were reduced in active smokers and ex-smokers as compared to non-smokers. Functionally, NK cell degranulation was almost absent in active smokers as compared to ex-smokers.

We have extensively analysed a number of parameters in relation to the patients' medical records, including age, smoking, COPD, FEV scores, levels of C-reactive protein, haemoglobin, and albumin, without finding any strong evidence for an association between trNK cells and these parameters (Fig. 2a below, not included in the manuscript). We did note a reduced expression of NKG2A specifically in CD69⁻ and CD69^{sp} CD56^{bright}CD16⁻ NK cells in current smokers, but that was not detectable in CD56^{dim}CD16⁺ or trNK cells (Fig. 2b below, not included in the manuscript). Furthermore, we noted that the increase in trNK cells in current

smokers was only significant in patients without COPD diagnosis, and not in patients with COPD, which is likely due to the lower power to detect such differences in the relatively few donors with COPD (Fig. 2c below, not included in the manuscript).

Together, due to the weak correlations between NK cell phenotypes and patient parameters, we decided to not add these data to the manuscript. In addition, we have decided to move the data about cigarette smoking to the supplements (in response to comments from reviewer #1).

Fig. 2: Patient parameters in relation to human lung NK cells. (a) Correlations of trNK cell frequencies and patient characteristics. **(b)** NKG2A expression on human lung NK cell subsets (conventional CD56^{dim}CD16⁺, CD16⁻CD69⁻CD49a⁻CD103⁻; CD16⁻CD69^{sp}, CD16⁻CD69⁺ co-expressing CD49a and/or CD103 (trNK)). Mean \pm SD is shown. **(c)** Frequency of trNK cells in patients with or without COPD in relation to the cigarette smoking status. Mean \pm SD is shown.

- *The authors describe in the result section that frequencies of NK cells were not affected by the medication taken by the patients (statins and corticoids). The authors should also address and provide data whether the major phenotypic changes reported in lung were affected by the medication taken by the patients.*

Answer:

In a separate analysis not included in the paper, we show that the NK cell composition in the lung was independent from medication (Fig. 3 below, not included in the manuscript), although the spread between the data for trNK cells was reduced in the statin-treated cohort as compared to patients not receiving any medication or receiving medication which was non-relevant for the study. Therefore, the presence of phenotypically different trNK cells in the lung is likely to

be affected by other factors than the medication, which could, however, not be addressed in the present manuscript.

Fig. 3: Frequencies of lung NK cell subsets are independent from medication. Patients have been grouped according to the medication (indicated in numbers), and frequencies of conventional CD56^{dim}CD16⁺ (left panel) and trNK cells (CD16⁻CD69⁺ NK cells co-expressing CD49a and/or CD103, right panel) were analysed. n=4-29. Mean ± SD is shown.

• *Were differences in NK Phenotypes observed depending on the cancer types? Was a correlation to prognosis detectable?*

Answer:

The vast majority of patients were diagnosed with adenocarcinoma (Fig. 4 below, not included in the manuscript). Furthermore, the large spread of data between patients in most groups regarding the frequency of conventional NK cells and trNK cells limits the possibility to compare different groups. Therefore, any differences between groups are inconclusive at this point.

Fig. 4: Frequencies of lung NK cell subsets in correlation to different types of cancer. Patients have been grouped according to the type of cancer (indicated in numbers), and frequencies of conventional CD56^{dim}CD16⁺ (left panel) and trNK cells (CD16⁻CD69⁺ NK cells co-expressing CD49a and/or CD103, right panel) were analysed. n = 3-58. Mean ± SD is shown.

• *Fig 2: CD49a/CD103: does this mean and/or (as commented in the figure legend) or both molecules to be co-expressed? Please clarify*

Answer:

We understand that the original definition of trNK cells in the Figure was unclear and have revised the corresponding figures and text. In detail, for protein analyses, we have applied the Boolean gate “CD69⁺ AND (CD49a⁺ OR CD103⁺)”, thus including CD69⁺CD49a⁺CD103⁺, CD69⁺CD49a⁺CD103⁻ and CD69⁺CD49a⁻CD103⁺ NK cells, which we together denote as “CD49a⁺ and/or CD103⁺ NK cells”. We have clarified which NK cell subsets have been analysed throughout the revised manuscript.

• Fig 3: CD8+ TRM cells vs TEM how were these cells isolated? This should be included in the result section

Answer:

The data from the study on CD8+ T cells (Kumar et al., ref. 1 in manuscript) were derived from T cells in human lungs, which were mechanically and enzymatically processed, similar as for the tissue used in the present study.

For RNA sequencing of T cell subsets from spleen and lung tissue, CD69- and CD69+ T cells were isolated from CD3+CD4+ and CD3+CD8+ (CD45RA-CCR7-), respectively, with CD69- T cells representing TEM and CD69+ T cells representing TRM cells (Kumar et al.).

We have now added this information in the results section in the revised manuscript (page 11, line 238/239).

• Fig 4: Polyfunctional NK cells: Do the same NK cells kill and produce cytokines? Are these different subsets? Is TRAIL expressed? How does the Digestion of the tissue Change different marker Expression?

Answer:

Re-analysis of the data from Fig. 4d showed that co-expression of CD107a (surrogate marker for killing capacity) and cytokine production is NK cell subset-dependent (Fig. 5 below, not included in the manuscript). In detail, only CD16-CD56^{bright} NK cells lacking CD103 co-expressed CD107a, GM-CSF, and/or TNF. NK cells expressing CD103 were mainly either TNF-single positive or negative for any of the effector molecules. Hence, and in conclusion from Fig. 4d in the revised manuscript, it is likely that CD103- and CD103+ NK cells have different functional characteristics, with CD103- NK cells being more polyfunctional than CD103+ NK cells.

Fig. 5: Co-expression of effector molecules on/in distinct NK cell subsets in human lung. Human lung NK cells were stimulated with PMA/ionomycin for 4h and subsequently analysed for co-expression of CD107a, GM-CSF, and TNF. n = 2-3. Mean ± SD is shown.

All NK cell subsets from human lung including trNK cells expressed only very low RNA levels of *TNFSF10* (encoding TRAIL), and no signal for TRAIL could be detected at the protein level on fresh human lung NK cells that were or were not enzymatically treated.

In response to the reviewer’s question regarding the effect of enzyme digestion on marker expression, we analysed expression levels of diverse markers on NK cells from human lung tissue and human tonsils which were or were not enzyme-treated (Fig. 6 below, not included in the manuscript). The data show that enzyme digestion resulted in decreased levels of CXCR6

on trNK cells in lung and NK cells in tonsil. In contrast, levels of CD45 were increased, and data for other markers were inconclusive. Importantly, we believe that it is fair to assume that the enzyme digestion affected all subsets which are compared in our study similarly, eliminating the effect of enzyme digestion on differences between NK cell subsets in lung.

Fig. 6: Surface receptor expression on NK cells from untreated or enzyme-treated human tissue. (a) Analysis of receptor expression on human lung trNK cells (n=3) and (b) on human tonsil NK cells (n=1).

• *Figure 5: the in vitro stimulation with IL-15 does not add much novel information (compared to ref 27) and appears out of context.*

Answer:

We agree with the reviewer that showing that IL-15 results in upregulation of CD49a on NK cells is not entirely novel, since it was previously demonstrated for NK cells in peripheral blood (Hydes et al., ref. 31 in the manuscript). However, we show here for the first time that these *de novo* CD49a⁺ lung NK cells display higher activation levels compared to CD49a⁻ NK cells when stimulating sorted CD16⁺ and CD16⁻CD49a⁻ NK cells with IL-15. The aim of the analysis, however, was primarily made to highlight the fact that these newly emerged CD49a⁺ NK cells might confound results when analysing total CD49a⁺ NK cells after IL-15 stimulation, unless defined populations of sorted CD49a⁺ and CD49a⁻ NK cells are used for the assays (as was done in the current manuscript). That said, the IL-15-induced CD49a expression might aid NK cells in establishing tissue-residency, e.g. upon viral infection in the lung which induces local IL-15 production, as has previously been suggested for tissue-resident T cells. We do however understand the reviewer’s opinion that the data appear out of context. Therefore, we moved former Fig. 5 to the supplements (new Fig. S4), slightly revised the figure and the text, and added a clear justification in the respective results section of the manuscript (page 15, line 319-324).

Minor comments

1. *Abstract: polyfunctional: please spell out the functional characteristics, this could help to clearly state novelty of the study and the take home message*

Answer:

We have now spelled out the functional characteristics regarding polyfunctionality in the abstract, which indeed clarifies the intention of the study.

2. *Abstract: last sentence: unique part..unclear what the authors mean with this*

Answer:

Based on our analyses, we demonstrate that human lung trNK cells are distinct both from non-trNK cells (e.g., CD16⁻CD69⁻CD49a⁻CD103⁻, or CD16⁺ NK cells) within the lung, but also distinct from trNK cells in other organs such as the bone marrow. Furthermore, lung trNK cells are different from lung CD8⁺ T_{RM} cells. We have revised the text in the abstract accordingly.

3. *Page 6, 108: non-NK ILC include how defined in text,*

Answer:

We have now added the respective information in the text (ILCs were defined as live CD45⁺CD14⁻CD19⁻CD3⁻CD16⁻NKG2A⁻KIR⁻CD69⁺CD127⁺) (page 7, line 124/125).

4. *Figure 2. It is explained in the result section that protein expression of Eomes is highest in trNK cells. However, it seems that mean value is same as in CD69sp NK cells.*

Answer:

The reviewer is correct in his/her statement that the levels of Eomes expression are not significantly different between trNK cells and CD69sp NK cells. We have revised the text accordingly (page 11, lines 220/221).

5. *Figure 4. clarify in the figure legend what n.a. means for CD49a on the graphs f and e.*

Answer:

We have revised the figure legend according to the reviewer's comment (n.a. = not analyzed).

Reviewers' comments:

Reviewer #1 (Remarks to the Author):

The authors have made extensive changes throughout the manuscript in the definition of cell populations as suggested.

However the manuscript is still very complicated to read and inconsistent in several aspects.

For example in the abstract there are several claims which are not supported by data or not specific for the cells described.

-trNK cells are still described as co-expressing CD103 and/or CD49a despite the fact that the authors have redefined the cells throughout the paper.

-CXCR6 expression is described as a characteristic, but in fact the differential expression is very minimal (on a linear axis) with an extensive spread in data.

- the upregulation after IL-15 of Ki67, perforin, granzymeB, CCL5 is not specific for these cells, but is for CD16-CD49a-as well and in fact for any NK cell population.

I still have problems with the following points:

Gating strategy Fig S1a.

in the CD69 vs CD16 plot the gate separating CD16pos and CD16neg cells is still set rather arbitrarily, why not gating CD69+ and CD69- cells first to define tr cells?

The gating strategy used to sort the cell populations for RNAseq is not clear and incomplete. For example CD56dim,CD16+, NKG2A+ cannot be deduced from the gates used. Is CD56 used to sort populations for RNAseq or not? (not described)

It is still confusing that definition of populations for the RNAseq experiments is different from the definition of cells in figure 2d-h in flow cytometry experiments confirming protein expression.

In figure 4b+c the definition is different again. (CD49a+ vs-)

Discussion:

I do not see protein data on MIP-1a (discussion line 385)

In conclusion, definition of cell populations that are not consistent, and preclude conclusions about these cells and their uniqueness.

Reviewer #2 (Remarks to the Author):

My comments were addressed very well in the Point to Point response. I would like to ask the authors to also include the Information that was provided for me as reviewer in the manuscript (eg the Impact of Treatment). This would further strengthen the manuscript.

Reviewers' comments:

Reviewer #1 (Remarks to the Author):

The authors have made extensive changes throughout the manuscript in the definition of cell populations as suggested.

Reply: We are happy to hear that the reviewer appreciates our efforts in responding to the previous suggestions by the referees.

However the manuscript is still very complicated to read and inconsistent in several aspects. For example in the abstract there are several claims which are not supported by data or not specific for the cells described.

Reply: We thank the reviewer for highlighting these concerns. We have now extensively rewritten the abstract and parts of the main text as well as edited the figures to bring full clarity to the issues raised. We feel that the present re-revised version is much improved in terms of being easier to read and in consistency. Furthermore, all claims that are made are supported by data for the specific cells described.

-trNK cells are still described as co-expressing CD103 and/or CD49a despite the fact that the authors have redefined the cells throughout the paper.

Reply: In the re-revised version, we have removed the description of trNK cells as "CD69⁺ cells co-expressing CD49a and/or CD103" from the abstract. Instead, to bring full clarity, we state that we identify CD69⁺ subsets of NK cells with hallmarks of tissue-residency, including expression of CD49a and CD103. In the results section of the re-revised manuscript, we have specified how the subsets were defined; i.e., CD69⁺CD49a⁻CD103⁻, CD69⁺CD49a⁺CD103⁻, and CD69⁺CD49a⁺CD103⁺ NK cells. We have attempted to use as clear definitions as possible to define specific NK cell subsets throughout the paper. Indeed, the revised definitions substantially enhance the clarity of the manuscript. They also facilitate the comparison between distinct defined subsets with respect to, e.g., protein and RNA expression.

Based on the strong co-expression of CD49a and CD103 as demonstrated in Fig. 1c, and since our data show that more than 85% of trNK cells express CD49a we have, however, adopted a simplified definition of trNK cells by virtue of CD49a expression alone for functional assays where cell numbers were too limited for specific subset analysis.

-CXCR6 expression is described as a characteristic, but in fact the differential expression is very minimal (on a linear axis) with an extensive spread in data.

Reply: We agree with the reviewer that differences in expression of CXCR6 is not striking when comparing the CD69⁻ CD16⁻ NK cell subset with CD69⁺ CD16⁻ trNK cell subsets. In line with the suggestion by the reviewer, we have therefore removed the description of CXCR6 as a defining feature for these cells from the abstract. We would, however, like to point out that CXCR6 was expressed by all CD69⁺ CD16⁻ NK cell subsets, all with clearly higher RNA levels than the CD69⁻ CD16⁻ NK cell subset (Fig. S4b).

- the upregulation after IL-15 of Ki67, perforin, granzymeB, CCL5 is not specific for these cells, but is for CD16-CD49a-as well and in fact for any NK cell population.

Reply: The reviewer is correct in pointing out that also CD49a⁻ CD16⁻ NK cells responded well to stimulation with IL-15. In the re-revised version of the abstract, we have clarified that the NK cell subsets with a tissue-resident phenotype responded to stimulation at largely similar levels as CD49a⁻ CD16⁻ NK cells. The main observation in the functional analyses of

the cells (Fig. 4), was that CD49a⁺ CD16⁻ NK cells were responsive *per se* to stimulation. Noteworthy, although most NK cells are capable of responding to IL-15, there are clear differences between different subsets, as exemplified by the relatively modest increase in Ki67 expression by CD56^{dim} CD16⁺ NK cells compared to the robust induction in the CD49a⁺ and CD49a⁻ CD16⁻ NK cell subsets.

I still have problems with the following points:

Gating strategy Fig S1a. In the CD69 vs CD16 plot the gate separating CD16pos and CD16neg cells is still set rather arbitrarily, why not gating CD69+ and CD69- cells first to define tr cells?

Reply: We understand the concern of the reviewer regarding the gating strategy described in Fig. S1a, with respect to CD16⁺ and CD16⁻ NK cells; in particular, since the axis of the plot referred to was compressed. The axes of the plot provided in Fig S1 showing the gating strategy have now been re-transformed to better illustrate the relatively distinct separation of CD16⁺ and CD16⁻ cells. It should also be noted, however, that in most samples the separation in CD16 expression was more distinct, e.g., as shown in Fig. S1d.

Since CD69 alone is not sufficient to clearly identify trNK cells, and since the vast majority of NK cells expressing hallmark markers of tissue-residency, e.g., CD49a and CD103, were confined to CD16⁻ NK cells, we felt that it was more logic to start off with the division of NK cells into CD16⁻ and CD16⁺ subpopulations. The division of NK cell cells into CD16⁻ and CD16⁺ subpopulations is also, subsequently, used throughout the paper. Moreover, the division of CD16⁻ and CD16⁺ NK cells separates the two major NK cell populations in humans and is commonly used in human NK cell analysis. As such, we feel that keeping the initial division of CD16⁻ and CD16⁺ NK cells is very well motivated. That said, we do provide data, as previously suggested by the reviewer, showing the contribution of NK cell subsets defined by expression of CD16, CD69, CD49a and CD103 (Fig. 1g), as well as the expression of CD16 on NK cell subsets defined by the expression of CD69, CD49a and CD103 (Fig. 1d, e). Taken together, we feel that the provided data, at present, provide a clear overview of the overall distribution of NK cell subsets in the human lung, as well as detailed insights into NK cell subsets expressing markers and other features associated with tissue-residency.

The gating strategy used to sort the cell populations for RNAseq is not clear and incomplete. For example CD56dim,CD16+, NKG2A+ cannot be deduced from the gates used. Is CD56 used to sort populations for RNAseq or not? (not described)

Reply: We apologize for not including the sorting strategy in a separate figure. In the re-revised manuscript, we now provide the gating strategy employed for the cells analysed by RNA sequencing (new Fig. S1f). Briefly, we first gated on single live CD45⁺CD14⁻CD19⁻CD3⁻ lymphocytes, followed by separate gates on CD56⁺CD16⁻ and CD56⁺CD16⁺ NK cells. Among the CD56⁺CD16⁻ NK cells, we subsequently gated on CD69⁻NKG2A⁺ NK cells and CD69⁺NKG2A⁺ NK cells. Within the CD69⁺NKG2A⁺CD16⁻ NK cell subset, we then sorted CD49a⁺CD103⁺, CD49a⁺CD103⁻, and CD49a⁻CD103⁻ NK cells. The rationale for including NKG2A as a criterion for the CD16⁻ NK cell subsets was to minimize the possibility of cross-contamination by CD56^{dim} NK cells that have downregulated CD16 expression (re-revised Fig. 2a, b, see also comment below). Indeed, analysis of the sorted NKG2A⁺CD16⁻ NK cell subsets used for RNA sequencing analysis revealed that ≥99% of the cells lacked expression of CD57 (Fig. S1g) and, hence, were unlikely to be affected by any potential downregulation of CD16.

To sort CD56⁺CD57⁻NKG2A⁺ cells out of the CD16⁺ NK cell population, we gated on CD57⁻NKG2A⁺ NK cells within the CD56⁺CD16⁺ population, as shown in Fig. S1f.

It is still confusing that definition of populations for the RNAseq experiments is different from the definition of cells in figure 2d-h in flow cytometry experiments confirming protein expression.

In figure 4b+c the definition is different again. (CD49a+ vs-)

Reply: The reviewer raises a good point. In order to facilitate comparison, we have re-analyzed protein data. We have removed the term “CD69⁺CD49a⁺ and/or CD103⁺”, and instead provide data on the individual subsets of CD69⁺CD49a⁻CD103⁻, CD69⁺CD49a⁺CD103⁻, and CD69⁺CD49a⁺CD103⁺ CD16⁻ NK cells (Fig. 2a-h). As previously pointed out by the reviewer, a subset of CD69⁻ CD16⁻ NK cells expressed CD57 (Fig. 2a, b), which is normally only observed on CD56^{dim}CD16⁺ NK cells. Hence, there was a possibility that the CD69⁻ CD16⁻ subset contained CD56^{dim} NK cells with downregulated CD16 expression. To further facilitate the comparison, and to minimize the potential contribution by CD56^{dim} NK cells with reduced CD16 expression, we therefore gated on CD69⁻CD57⁻NKG2A⁺ CD16⁻ NK cells for comparing protein expression of CXCR3, CXCR6, perforin, Eomes, and T-bet between this subset and the three CD69⁺ CD16⁻ NK cell subsets (re-revised Fig. 2c-h).

In relation to results presented in Fig. 4b-c, we did not include anti-CD103 in the staining panel. This in order to be able analyse a larger number of cytokines/chemokines. However, given that the vast majority of CD103⁺ NK cells were confined within the CD49a⁺ CD16⁻ NK cell subset (as seen in Fig. 1a-c), the CD49a⁺ NK cells in the analysis will to a very large extent also express CD103. As such, the functional data on CD49a⁺ NK cells are representative of the vast majority of lung NK cells with a tissue-resident phenotype, allowing us to draw the conclusion that these cells are functionally competent. This finding is also supported by the data in Figure 4d-f, which further dissects the functional responses by CD49a⁺ NK cells. In order to avoid any confusion in the manuscript due to identification of trNK cell subsets at RNA and protein levels, we now more in detail describe the trNK cells used in functional analyses.

Discussion:

I do not see protein data on MIP-1a (discussion line 385)

Reply: We thank the reviewer for spotting this mistake. We have analysed protein expression of MIP-1 β , not MIP-1 α , and have changed the text accordingly.

In conclusion, definition of cell populations that are not consistent, and preclude conclusions about these cells and their uniqueness.

Reply: We understand the concerns raised above with respect to ambitions to bring further clarity to the results presented in the manuscript, in particular the handling and description of the different subsets throughout the manuscript. During the two revisions, we have made a large effort to more accurately define the subsets analysed. Although the definition of NK cells with a tissue-resident phenotype varies slightly between figures due to limitations in cell numbers or technical aspects, all analyses include comparisons between CD49a⁺ NK cells, representing the vast majority of trNK cells, and CD49a⁻ NK cells, allowing us to draw conclusions about these subsets. Furthermore, in most analyses we further dissect the trNK cells with respect to both CD49a and CD103 expression. First, our data show that there are distinct subsets of CD16⁻ NK cells in the human lung, including CD69⁺CD49a⁻CD103⁻, CD69⁺CD49a⁺CD103⁻, CD69⁺CD49a⁺CD103⁺ and CD69⁻ subsets. Secondly, at the RNA level, we show that CD69⁺CD49a⁺CD103⁺ and CD69⁺CD49a⁺CD103⁻ CD16⁻ NK cells are clearly distinct from both CD69⁺CD49a⁻CD103⁻ (CD69sp) and CD69⁻ CD16⁻ NK cells, and express hallmarks of tissue-resident lymphocytes. The RNAseq analysis also defined gene expression signatures that were shared with trNK cells in bone marrow and CD8⁺ T_{RM} cells in lung, further supporting the notion that the CD69⁺CD49a⁺ NK cell subsets described above are indeed tissue-resident lymphocyte. Finally, we show that the NK cells with a

tissue-resident phenotype are functional, indicating that they readily could mount efficient responses to, e.g., viral infections in the lung.

Reviewer #2 (Remarks to the Author):

My comments were addressed very well in the Point to Point response. I would like to ask the authors to also include the Information that was provided for me as reviewer in the manuscript (eg the Impact of Treatment). This would further strengthen the manuscript.

We very much appreciate this reviewer's positive response and also agree that, in fact, the data covering clinical information would be interesting for the readers of the article after publication. Hence, we now provide the relevant clinical information in a new supplementary figure (Fig. S2a-c), which is also cited in the text of the manuscript.

REVIEWERS' COMMENTS:

Reviewer #1 (Remarks to the Author):

In the re-revised manuscript the authors have made the text much more readable. The gating of cells has now been consistently indicated and the nomenclature has been simplified. The addition of NKG2A and CD57 to define cells in the protein experiment is a relevant improvement.

I still have several remarks:

- Figure 2h, the + and - signs below the graph are not correct I think. (for example, there are 2 populations with all minus signs, this is very different form all other figures)
- In the heatmaps, the genes mentioned in the text have been accentuated, is that consistently performed? For example, I could not find ZNF331 in figure 2 and supp figures.
- as a service to the reader the description of the data presented in fig 3d, can be supplemented, either with the mentioning of the colors in the text, or accentuation of the mentioned genes as in fig 2, or including in the figure some indication what the colors mean (it is included in the legend, I realize, but it is quite a puzzle).
- one main question I have is about the differences between the CD49A+CD103 pos and neg cells. In most assays, as well as in differential expression data, these 2 populations are almost identical, except in fig 4d, where the two populations are analyzed separately, the CD103+ cells do not produce CD107 and GM-CSF, but they make TNFa. This is surprising, is it not a technical issue as there are very few cells?
- as for CXCR6, the statement has been deleted from the abstract, because the data are not very strong, but in the discussion the focus is still on the CXCR6 data, while the results for CXCR3 are more convincing in my opinion.

Response to reviewer comments

Below we provide a point-by-point response to the comments from the reviewer.

Reviewer #1 (Remarks to the Author):

In the re-revised manuscript the authors have made the text much more readable. The gating of cells has now been consistently indicated and the nomenclature has been simplified. The addition of NKG2A and CD57 to define cells in the protein experiment is a relevant improvement.

We thank the reviewer for the current and past careful scrutiny of the manuscript and were happy to hear that our revisions have made the text clearer.

- Figure 2h, the + and - signs below the graph are not correct I think. (for example, there are 2 populations with all minus signs, this is very different from all other figures)

We thank the reviewer for identifying this and have revised the figure accordingly.

- In the heatmaps, the genes mentioned in the text have been accentuated, is that consistently performed? For example, I could not find ZNF331 in figure 2 and supp figures.

We have to the best of our knowledge tried to accentuate potentially interesting genes, *e.g.*, transcription factors that might be involved in regulating trNK cells. The transcription factor ZNF331 is indeed upregulated in CD69⁺CD49a⁺CD103⁺CD16⁻ NK cells compared to CD69⁻CD16⁻ NK cells (*padj*=0.014, *log2FC*=1.26), but was not shown in the heatmaps due to the fact that we used a cut-off for *padj*<0.01 to reduce the number of genes displayed. A summary of all differentially expressed genes with *padj*<0.05, including ZNF331, is however provided in supplementary data 1 in the manuscript.

- as a service to the reader the description of the data presented in fig 3d, can be supplemented, either with the mentioning of the colors in the text, or accentuation of the mentioned genes as in fig 2, or including in the figure some indication what the colors mean (it is included in the legend, I realize, but it is quite a puzzle).

We have now added an extra legend to Fig. 3d and also mention the colours in the main results section. We hope that this facilitates the reading of the paper and further contributes to the clarity of the manuscript.

- one main question I have is about the differences between the CD49A⁺CD103⁺ pos and neg cells. In most assays, as well as in differential expression data, these 2 populations are almost identical, except in fig 4d, where the two populations are analyzed separately, the CD103⁺ cells do not produce CD107 and GM-CSF, but they make TNF α . This is surprising, is it not a technical issue as there are very few cells?

The reviewer is correct in that these two populations are overall similar at the RNA and protein level, although some differences do exist at both protein level (*e.g.*, Eomes), and RNA level (*e.g.*, *SELL* and *IRAK3*). The fact that both subsets respond equally well in terms of production of TNF in fact argues that the lower level of degranulation and GM-CSF production by CD49a⁺CD103⁺ NK cells compared to CD49a⁺CD103⁻ NK cells is not due to a technical issue. The difference was also reproducible and consistent between donors and experiments, further supporting that this finding is not merely caused by technical limitations in the assay. We would also like to mention that in all assays, we include sufficient cell numbers to reliably gate on responding cells. The lower degranulation (measured by CD107a cell surface expression) is furthermore in line with the lower RNA expression of *LAMP-1*

(encoding CD107a) by CD49a⁺CD103⁺ NK cells compared to CD49a⁺CD103⁻ NK cells (as shown in Fig. 4a).

- as for CXCR6, the statement has been deleted from the abstract, because the data are not very strong, but in the discussion the focus is still on the CXCR6 data, while the results for CXCR3 are more convincing in my opinion.

We agree with the reviewer in that the results for CXCR3 are as important as the data for CXCR6 and have therefore extended the discussion regarding CXCR3. However, CXCR6 is a well-described and established marker for trNK cells at least in lymphoid tissue and liver, and, as we mention in our discussion, the ligand for CXCR6, CXCL16, can be produced in the lung. CXCR6 was used to identify tissue-resident NK cells in bone marrow for RNA-seq analyses (Melsen *et al.*, 2018), which we used for comparison to our datasets. Furthermore, although CXCR6 protein data were less strong in our study, the gene *CXCR6* was differentially expressed in the lung and bone marrow as well as in lung and spleen CD8⁺ T_{RM} (Fig. 3a-c), with particularly higher expression on trNK cells as compared to CD69⁻ NK cells (Fig. S4b). We therefore feel that it is motivated to keep the present discussion regarding the role of CXCR6.